# Genomic consequences of residual recombination in a hybrid apomictic hickory complex

Wei-Ping Zhang[1,2,3], Sylvain Glémin[3,4], Xiao-Xu Pang[2], Ming Kang ●[1] ✉, Da-Yong Zhang ●[2] ✉, Martin Lascoux ●[3] ✉ & Wei-Ning Bai ●[2] ✉

Apomixis, a form of clonal asexual reproduction in plants, is often accompanied by residual sex, yet its genomic consequences remain poorly understood. Here, we assembled a haplotype-resolved genome of *Carya hunanensis* and analyzed whole-genome resequencing data from 195 adults and 180 mature embryos across four hickory species, representing a hybrid apomictic complex with both sexual and asexual lineages. We find apomictic species exhibited genomic signatures of clonality, notably loss of heterozygosity (LOH), suggesting recombination induced by rare sexual events. Despite harboring more heterozygous deleterious variants, apomictic adults showed lower realized mutation loads, particularly in hybrid *C. hunanensis*, whose apomictic haplotype disproportionately carried deleterious alleles. Remarkably, rare embryos from apomicts underwent recombination-mediated LOH, exposing deleterious mutations to selection. These findings reveal the genetic cost of residual sex, while also indicating its role in generating novel genotypes, supported by close relatedness among adult apomicts. Our study provides a unique genomic snapshot of how residual sex and recombination mitigate mutation accumulation and potentially facilitate clonal maintenance in natural asexual systems.

Sexual reproduction, characterized by meiosis and fertilization, dominates in animals and flowering plants. While facilitating genetic recombination and biparental inheritance, it also comes with substantial evolutionary costs, including energetic demands, disruption of co-adapted gene complexes, and the halving of parental genome transmission[1,2]. In contrast, obligate asexuality avoids these burdens[3], but is associated with other disadvantages. Transitions between the two modes of reproduction are rarely abrupt and often involve intermediate reproductive modes that combine asexuality with some recombination[4,5]. This residual recombination can lead to loss of heterozygosity (LOH), exposing recessive deleterious alleles and may reducing long-term viability[6,7]. Despite this, some asexual lineages

have evolved mechanisms to mitigate genetic load and maintain heterozygosity[8–10]. However, these mechanisms are often system-specific, underscoring the importance of investigating diverse asexual reproductive strategies.

Apomixis is a form of asexual reproduction in plants, in which seeds are produced without meiosis (or its avoidance, apomeiosis) and fertilization[11–13]. It is classified as either gametophytic or sporophytic, depending on whether unreduced embryo sacs originate from gametic or somatic cells[14]. Most apomictic taxa are facultative, combining sexual and asexual modes and often retaining features like partial meiosis or pollination-induced endosperm[15,16]. As a result, maternal plants can sometimes produce both sexually and asexually derived

[1]State Key Laboratory of Plant Diversity and Specialty Crops, South China Botanical Garden, Chinese Academy of Sciences, Guangzhou, China. [2]Ministry of Education Key Laboratory for Biodiversity Science and Ecological Engineering, College of Life Sciences, Beijing Normal University, Beijing, China. [3]Department of Ecology and Genetics, Evolutionary Biology Centre, Uppsala University, Uppsala, Sweden. [4]Université de Rennes, CNRS, ECOBIO (Ecosystémes, Biodiversité Evolution), Rennes, France. ✉e-mail: mingkang@scbg.ac.cn; zhangdy@bnu.edu.cn; martin.lascoux@ebc.uu.se; baiwn@bnu.edu.cn

seeds within a single generation, even within a single fruit in multi-ovulate species[5,17], highlighting the complexity and variability of apomictic reproduction. Polyploidy is prevalent among apomictic taxa of hybrid origin, a combination thought to confer genome buffering and increase environmental resilience[18–20]. In contrast, diploid hybrid apomicts, which have been reported rarely in *Borella*[21,22] and some *Citrus* species[23,24], may be more vulnerable. Despite its adaptive advantages such as reproductive assurance, local genotype fixation, and increased colonization ability, apomixis remains taxonomically rare, occurring in fewer than 0.15% of flowering plant species (roughly 400 out of 300,000)[16,25,26].

This paradox has led to hypotheses about evolutionary constraints on apomictic lineages[27]. One of the main hypotheses points to its developmental origin: apomixis may arise from incomplete or unstable reprogramming of the conserved sexual pathway, making it prone to disruption[28–30]. Theoretical models, including Muller's ratchet and the Hill–Robertson effect, further predict that reduced recombination in asexuals weakens purifying selection, allowing mildly deleterious mutations to accumulate over time and ultimately reducing evolutionary potential[31–33]. However, empirical comparisons between apomictic and sexual lineages have yielded mixed results: some asexuals show increased genetic load[22], while others do not or even exhibit genome homogenization[34,35]. Moreover, although "a bit of sex" might mitigate or prevent Muller's ratchet[36,37], the influence of facultative apomixis is often overlooked in genomic studies[22,38]. If occasional sexual reproduction helps maintain apomictic lineages, one would expect to find multiple distinct but closely related clonal lineages (e.g., parent-offspring or sibling-like structure) within a species, an observation that remains underexplored. There is therefore a need to clarify the genomic consequences of residual recombination in apomicts, including that arising from rare sexual reproduction or historical hybridization, and to assess how such processes influence lineage persistence.

A group of closely related hickory species, *Carya cathayensis*, *C. dabieshanensis*, *C. hunanensis*, and *C. tonkinensis* (Supplementary Fig. 1), offers a promising model for studying natural apomixis in diploid plants. All four species are endemic to southern China and valued for their pecan-like edible kernels, yet they exhibit allopatric distributions and remain largely confined to specific regions[39–41]. *Carya cathayensis* was first shown to exhibit apomixis through detailed embryological analyses showing nucellar embryogenesis[42], and supported by controlled pollination experiments showing seed development without paternal contribution[43]. Polyembryony, a trait associated with sporophytic apomixis[44,45], has been reported in both *C. cathayensis* and *C. hunanensis*[46]. Recent phylogenomic analyses indicate that *C. hunanensis* is a hybrid lineage derived from *C. cathayensis*, *C. dabieshanensis*, and *C. tonkinensis*[47], a finding supported by its intermediate husk morphology (Supplementary Fig. 1). Two distinct morphotypes, TD and YL, have been identified in allopatric populations of *C. hunanensis*, suggesting potential cryptic lineage diversification. Although apomixis and hybrid origin appear to coexist in this complex, the facultative nature of apomixis remains unconfirmed due to the absence of population-scale genomic evidence.

In this study, we assembled a haplotype-resolved, chromosome-level reference genome for the diploid *C. hunanensis* (2n = 2x = 32; Fig. 1 and Supplementary Figs. 2–3), and conducted whole-genome resequencing of 131 newly collected leaf samples from adult trees, together with 64 publicly available genomes[48], totaling 195 individuals across the four *Carya* species (Supplementary Data 1). We also generated a novel dataset by sequencing 180 mature embryos from three apomictic species (Supplementary Data 2). Using this integrated genomic dataset covering both vegetative and reproductive phases, we provide robust evidence for both interspecific hybridization and facultative apomixis within this species complex, and address the following key questions: 1) Does apomixis increase the accumulation of deleterious mutations relative to sexual relatives? 2) What is the relative contribution of apomictic and sexual parental lineages to genetic load in hybrid apomicts? 3) How does recombination-mediated LOH shape deleterious mutation patterns and affect lineage persistence?

## Results

### Haplotype-resolved genome assembly of *Carya hunanensis* and its hybrid origin supported by population structure

To generate a high-quality reference genome for the putative hybrid *C. hunanensis*, we produced 153.27 Gb (~218× coverage) of multi-platform sequencing data using Illumina, PacBio, and Hi-C technologies. The final assembly totaled 1.34 Gb (contig N50 = 11.31 Mb; scaffold N50 = 43.81 Mb) and exhibited high contiguity, coverage, and accuracy, with 97.13% of sequences anchored to 32 pseudochromosomes (Fig. 1; Supplementary Figs. 4–5 and Supplementary Table 1). Approximately 50.33% of the genome consisted of repetitive elements and a total of 78,799 protein-coding genes were predicted in the initial, unphased assembly (Supplementary Tables 1–2). This elevated gene count is consistent with the nearly doubled genome size (1.34 Gb) relative to the estimated haploid genome size (701.66 Mb) based on 19-mer analysis. Together with a high heterozygosity rate of 2.27% (Supplementary Fig. 3), these results indicated that both divergent haplotypes were represented. Mapping patterns revealed contrasting alignment preferences among the three putative parental species: *C. cathayensis* and *C. dabieshanensis* aligned predominantly to the first 16 pseudochromosomes (96.59% and 96.77%, vs. ~31% to the latter), whereas *C. tonkinensis* displayed the reverse (97.36% to the latter 16 vs. 28.52% to the first). These results enabled the partitioning of the *C. hunanensis* genome into haplotype E (Chr1A–16A) and haplotype W (Chr1B–16B), which was further confirmed by SubPhaser[49] analysis (Fig. 1 and Supplementary Figs. 6–7). Gene prediction on the separated haplotypes yielded 38,635 and 39,070 genes (Supplementary Table 1), respectively, in line with the expected gene content of a diploid genome.

We analyzed whole-genome resequencing from 195 leaf samples (131 newly collected and 64 publicly available genomes) representing four *Carya* species in southern China: 55 *C. cathayensis*, 47 *C. dabieshanensis*, 57 *C. hunanensis* (TD and YL morphotypes), and 36 *C. tonkinensis* (Fig. 2a)[47,48]. The average sequencing depth ranged from 21.61× to 26.03×, with mapping to the *C. hunanensis* haplotype E genome (Supplementary Data 1). ADMIXTURE[50] analysis at optimal K = 6 identified six genetic clusters (Fig. 2b and Supplementary Fig. 8). *Carya hunanensis* separated into two morphotypes (TD and YL) along with two admixed individuals (Chudx01 and Chuzy01); *C. cathayensis* and *C. tonkinensis* each formed distinct clusters, while *C. dabieshanensis* comprised at least two primary subclusters. Notably at K = 2, where *C. cathayensis* and *C. dabieshanensis* are lumped into one cluster and *C. tonkinensis* formed a separate group, *C. hunanensis* displayed ~47% ancestry from the former group and ~53% from the latter. The YL morphotype remained admixed at K = 3–4 but formed an independent cluster by K = 5 (Fig. 2b). PCA corroborated these findings, with PC1 (35.07% variance) positioning *C. hunanensis* intermediately between the three putative parental species, and PC2 separating the TD and YL morphotypes (Fig. 2c). These findings were further supported by significant pairwise divergence ($F_{ST}$ and $d_{xy}$) among the five taxa (Supplementary Fig. 9), and HyDe[51] analysis confirming the hybrid origin of *C. hunanensis* (Z > 344, P = 0; ~41% ancestry derived from *C. cathayensis* and/or *C. dabieshanensis*, and ~59% from *C. tonkinensis*; Supplementary Data 3), with γ values close to 0.5, indicating limited hybridization events.

The chloroplast phylogeny of 196 individuals (including *C. illinoinensis* as the outgroup) resolved five clades corresponding to the three focal species and two *C. hunanensis* morphotypes (Supplementary Fig. 10). The TD morphotype and two admixed individuals clustered with *C. dabieshanensis*, while the YL morphotype formed a sister lineage within the same clade. Haplotype analysis revealed a single

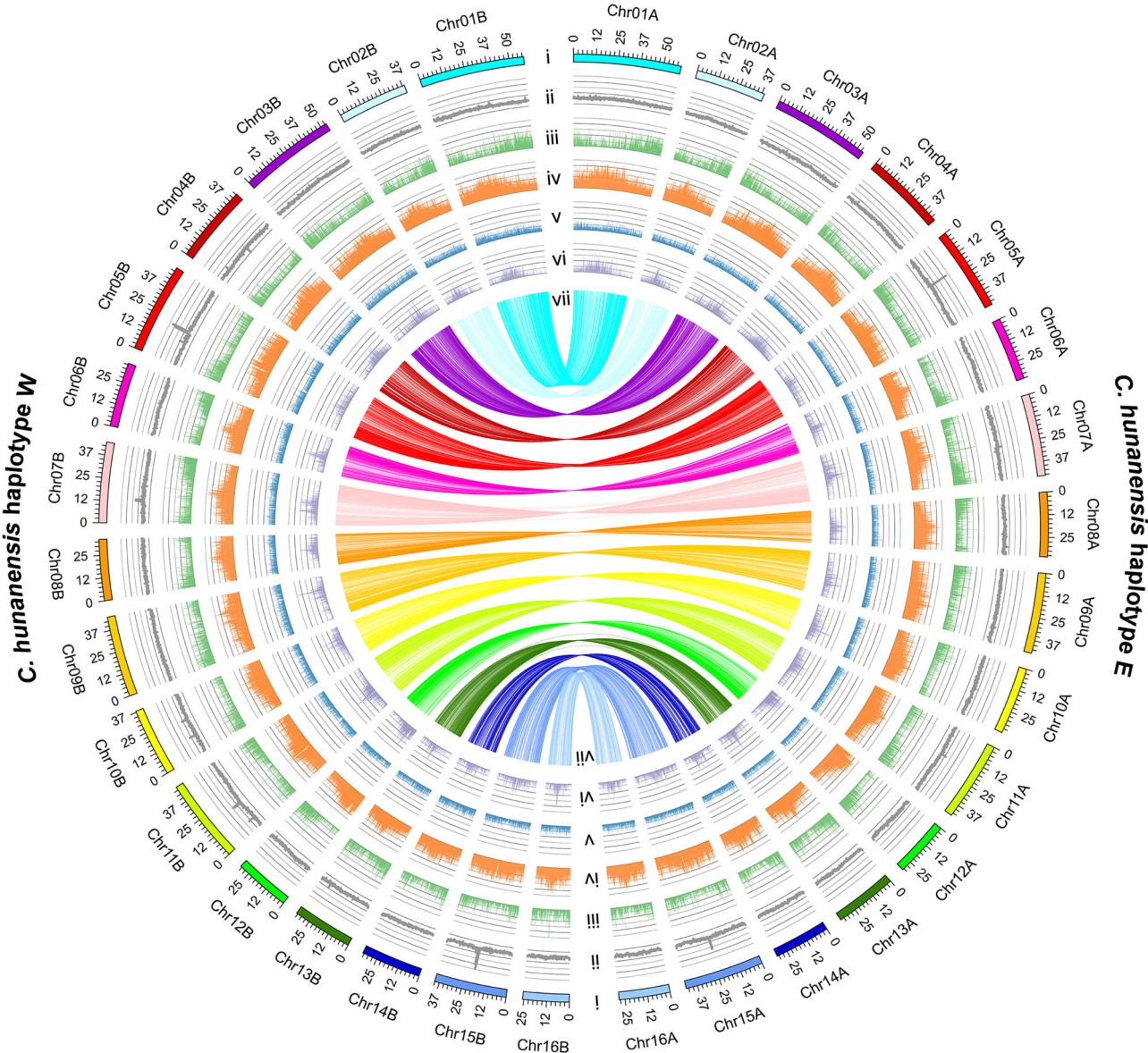

**Fig. 1 | Circos plot of haplotype-resolved genome of *Carya hunanensis*.** Different tracks (moving inward) represent: (i) 16 chromosome pairs, with ChrA and ChrB; (ii) GC content; (iii) gene density; (iv) transposable element (TE) density; (v) LTR-Copia density; (vi) LTR-Gypsy density, all in 500-kb stepping windows; (vii) syntenic blocks between homologous chromosomes from two genome haplotypes. The two haplotypes, designated as haplotype W and haplotype E, were named based on their inferred parental geographic origins from western and eastern China, respectively. Source data are provided as a Source Data file.

haplotype in all 55 sampled *C. cathayensis* individuals, likely reflecting its apomictic reproduction[42], two haplotypes in *C. dabieshanensis*, four in *C. hunanensis*, and greater diversity in *C. tonkinensis* (Fig. 2d). These results suggest that *C. hunanensis* likely inherited its maternal lineage from a variant of *C. dabieshanensis* (8–10 bp divergence), with *C. tonkinensis* as the paternal lineage (Fig. 2d and Supplementary Fig. 10).

### Polyembryony and genomic signatures of clonality and residual recombination in three apomictic hickory species

Our germination experiments with 120 mature seeds per taxa revealed polyembryony (a hallmark of sporophytic apomixis[44,45]) in *C. cathayensis* (34.26%), *C. dabieshanensis* (24.77%), and both morphotypes of *C. hunanensis* (>98%), whereas *C. tonkinensis* produced only single seedlings (Supplementary Fig. 11 and Supplementary Table 3). The widespread occurrence of polyembryony, particularly in the hybrid *C. hunanensis* where seeds yielded up to eight seedlings, strongly supports apomixis in these taxa.

Kinship analysis was performed on genome-wide SNP data from adult individuals comprehensively sampled to represent the full geographic distribution of each species. All 55 individuals of *C. cathayensis* formed a single clonal lineage, potentially reflecting obligate apomixis in all adult trees of this species (Fig. 3a). Similarly, 44 of the 47 individuals of *C. dabieshanensis* grouped into five clonal lineages (comprising 9, 21, 9, 2, and 3 individuals; Fig. 3b), and 55 of 57 individuals of *C. hunanensis* (including 44 of the TD morphotype and 11 of the YL morphotype; Fig. 3c) were grouped into two clonal lineages, all possibly derived from apomictic reproduction. Pairwise comparisons within *C. dabieshanensis*, involving the five clonal lineages and the three remaining individuals (Cdabmj02, Cdazf01, and Cdazf02), as well as within *C. hunanensis*, involving the two clonal lineages and the two remaining individuals (Chudx01 and Chuzy01), revealed first- to third-degree relationships, suggesting occasional sexual reproduction (Fig. 3b, c). In contrast, no clonality was detected in sexually

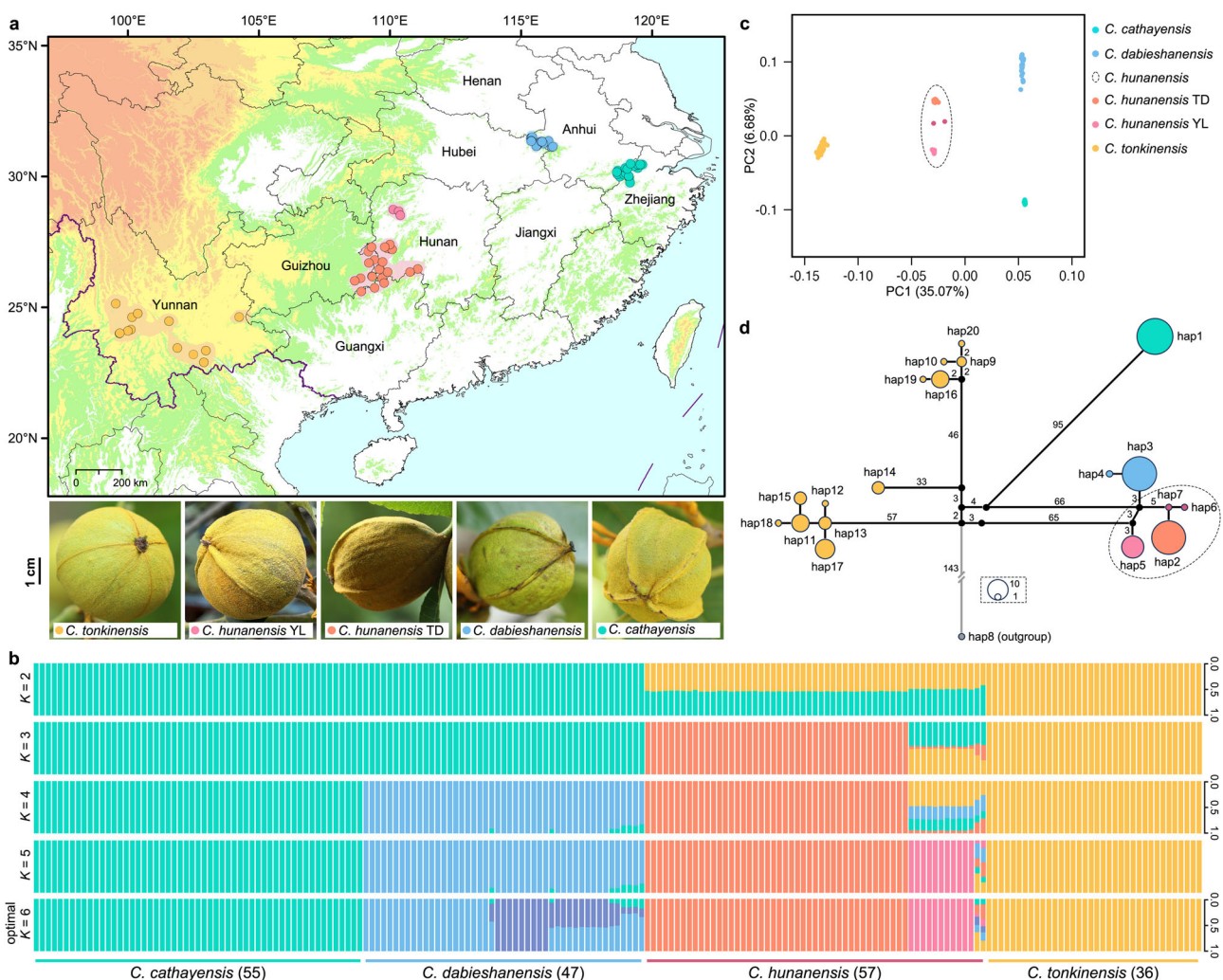

**Fig. 2 | Population structure and hybrid genomic variation in the apomictic hickory species complex. a** Sample distributions and representative fruits of *Carya cathayensis*, *C. dabieshanensis*, the TD morphotype of *C. hunanensis*, the YL morphotype of *C. hunanensis*, and *C. tonkinensis*. Shaded areas represent the extant distributions of each taxon in China, based on herbarium records and field surveys. Field observations revealed that the husks of *C. cathayensis* and *C. dabieshanensis* are winged from the apex to the base, *C. tonkinensis* husks lack prominent wings, and *C. hunanensis* husks are winged from the middle to the base. **b** Population structure of 195 adult individuals across four hickory species. Results of ADMIXTURE analysis for $K = 2–6$ are shown, with $K = 6$ inferred as the optimal cluster number. **c** PCA plot based on genetic covariance among 195 adult individuals of four hickory species, showing the first two principal components (PCs). **d** Chloroplast haplotype network for four hickory species, with the number of haplotypes and base differences displayed. Different colored lines or circles represent distinct hickory taxa. Source data are provided as a Source Data file.

reproducing *C. tonkinensis*, where most pairwise relationships (602/630) were more distant than third-degree (Supplementary Fig. 12).

The folded site frequency spectrum (SFS) analysis revealed distinct genetic signatures. In *C. tonkinensis*, the SFS exhibited a unimodal distribution characteristic of sexually reproducing species (Supplementary Fig. 13a). In contrast, *C. cathayensis* showed a bimodal distribution with a pronounced peak in the first and last bin (Supplementary Fig. 13b). Likewise, *C. dabieshanensis* and *C. hunanensis* exhibited multimodal SFS patterns, with the three largest clonal groups of *C. dabieshanensis* (21, 9, and 9 individuals) and both TD and YL morphotypes of *C. hunanensis* closely resembling the SFS of *C. cathayensis* (Supplementary Fig. 13c, d). These patterns indicate that apomixis leads to reduced genetic diversity and skewed allele frequency spectra.

Sexually reproducing individuals typically show variable and stochastic LOH patterns due to recurrent recombination and random mating. In contrast, several asexual lineages may exhibit shared LOH profiles within clonal groups, reflecting non-random and recurrent patterns along chromosomes. In principle, such patterns could arise

from mitotic DNA double-strand break repair, which is generally expected to generate only very short and localized LOH tracts and is unlikely to occur simultaneously across multiple chromosomes[52], or from automixis, a modified form of meiosis predicted to produce genome-wide but structured LOH with positional biases around centromeres or telomeres, although this process has not yet been rigorously confirmed in angiosperms[4,53]. Residual sexual reproduction within apomictic clones could also generate such patterns, as it would be expected to produce extended LOH tracts at variable chromosomal positions, and subsequent selfing or mating among closely related clonal lineages could give rise to novel LOH profiles distinct from those of the parental clones (illustrated schematically in Supplementary Fig. 14). To assess genomic LOH patterns, we analyzed heterozygosity in 50-kb sliding windows across the 16 chromosomes of 195 adults from four *Carya* species. Using a cutoff of <0.001, benchmarked against the sexual control *C. tonkinensis* (Supplementary Figs. 15–16), we found nearly identical LOH blocks in all *C. cathayensis* individuals and eight distinct LOH profiles in *C. dabieshanensis* that matched to its clonal groups (Supplementary Figs. 17–18), both may reflect

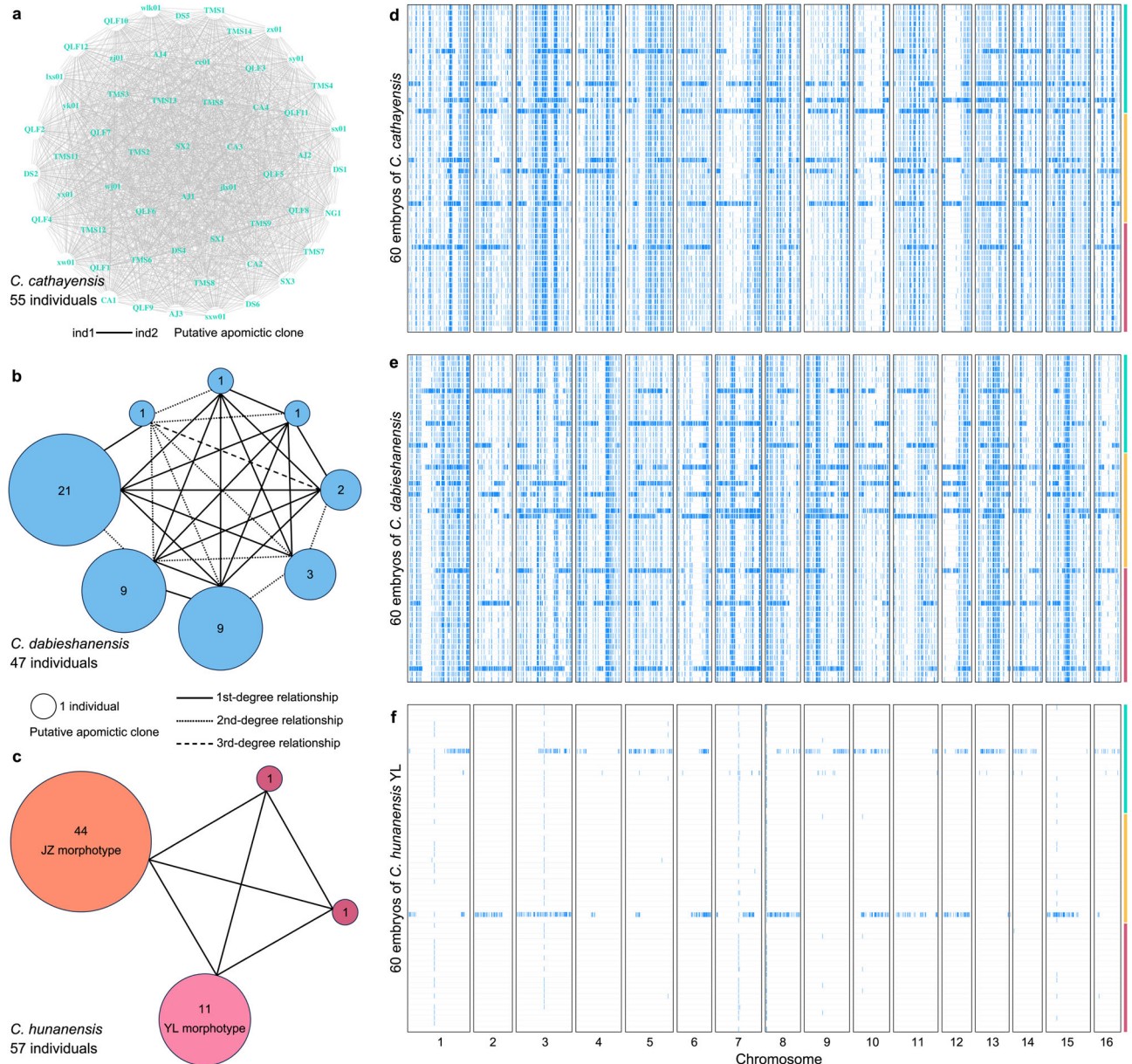

**Fig. 3 | Genomic signatures of apomixis in three hickory species, with sporadic sexual reproduction detected in mature embryos.** Kinship relationships among all sampled adult individuals of *Carya cathayensis* (**a**), *C. dabieshanensis* (**b**), and *C. hunanensis* (**c**). Lines link individual pairs identified as duplicates, representing putative apomictic clones. Each circle represents a clone, with its size proportional to the number of individuals it contains, and the count displayed at the center. Line formats denote different kinship degrees (first, second, or third). Circle colors and individual names correspond to taxa. **d**–**f** Loss of heterozygosity (LOH) distribution on 16 chromosomes in three putative apomictic *Carya* species (**d**: *C. cathayensis*, **e**:

*C. dabieshanensis*, and **f**: the YL morphotype of *C. hunanensis*). Data are based on low-depth resequencing of 60 mature embryos per species, sampled from three wild adult trees (18–21 embryos per tree, indicated by the three differently colored bars on the right). Genomic windows of 50-kb with heterozygosity density below 0.001 were classified as LOH regions and are shown as blue bars across the chromosome. The extended LOH segments likely reflect meiotic recombination events, indicating possible sexual reproduction. Source data are provided as a Source Data file.

recombination-derived LOH tracts subsequently fixed through clonal propagation. In *C. hunanensis*, LOH distributions were largely absent, likely due to its hybrid origin (Supplementary Fig. 19). *Carya tonkinensis* exhibited no shared LOH patterns, supporting its obligately sexual nature (Supplementary Fig. 20).

To assess whether new LOH could arise from residual sexual recombination, we performed low-coverage (10×) whole-genome sequencing of 180 mature embryos from three apomictic taxa, with 60 embryos per species sampled from three trees (Supplementary Data 2). Under the same threshold as above, only 8 (*C. cathayensis*), 11 (*C. dabieshanensis*), and 3 (*C. hunanensis*, YL morphotype) embryos

displayed large LOH segments that differed from those in most other embryos (Fig. 3d–f), indicative of rare recombination events during embryo development and plausibly arising from residual sexual reproduction. Applying a more stringent threshold of heterozygosity <0.0005 yielded consistent LOH patterns in both adult and embryo datasets (Methods; Supplementary Figs. 21–26).

## Comparison of genetic load between apomictic and sexual hickory species

To evaluate genetic diversity and selection efficacy among the four hickory species, we analyzed genome-wide SNP data from 195 adult

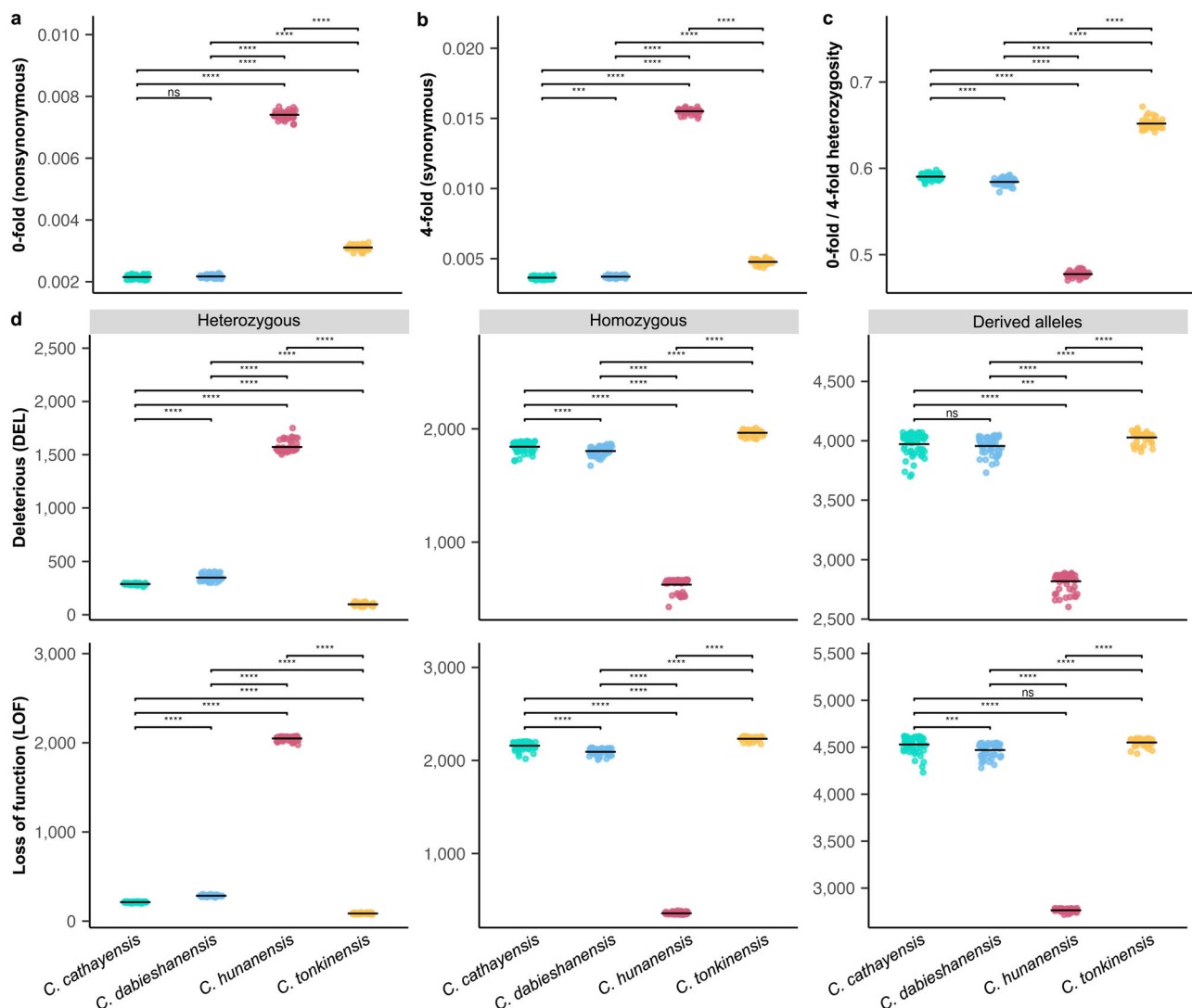

**Fig. 4 | Genetic load in adult individuals of apomictic versus sexual hickory species.** Nucleotide diversity at nonsynonymous (0-fold degenerate) sites (**a**) and synonymous (4-fold degenerate) sites (**b**). **c** Ratio of heterozygosity at 0-fold relative to 4-fold sites. **d** Deleterious mutation load in heterozygous, homozygous, and total derived alleles. All analyses are based on 195 adult individuals from four hickory species (55 *C. cathayensis*, 47 *C. dabieshanensis*, 57 *C. hunanensis* and 36 *C. tonkinensis*). Total derived allele counts were obtained by scoring each heterozygous genotype as one and each homozygous-derived genotype as two. Results are based on SIFT4G predictions using the *C. hunanensis* haplotype E reference genome (derived from apomictic lineages). Horizontal bars denote average values. Statistical significance between species was assessed using two-sided Welch's t-tests without multiple-testing correction. Asterisks indicate significance levels (ns, not significant; \**P* < 0.05; \*\**P* < 0.01; \*\*\**P* < 0.001). Source data are provided as a Source Data file.

individuals. As expected, the apomictic hybrid *C. hunanensis* exhibited the highest genome-wide nucleotide diversity ($\pi = 1.24 \times 10^{-2}$) and individual heterozygosity ($h = 1.48 \times 10^{-2}$), with similar values across its TD and YL morphotypes. This was followed by the sexually reproducing *C. tonkinensis* ($\pi = 9.62 \times 10^{-3}$; $h = 4.17 \times 10^{-3}$), while the apomictic species *C. dabieshanensis* ($\pi = 4.48 \times 10^{-3}$; $h = 3.84 \times 10^{-3}$) and *C. cathayensis* ($\pi = 3.40 \times 10^{-3}$; $h = 3.73 \times 10^{-3}$) had substantially lower values. Purifying selection efficacy was assessed using the ratio of heterozygosity at 0-fold (nonsynonymous) to 4-fold (synonymous) degenerate coding sites. Among the four species, *C. hunanensis* exhibited the highest absolute heterozygosity at both site classes (all $P < 2.2 \times 10^{-16}$; Fig. 4a, b), likely reflecting the retention of divergent parental haplotypes. However, it showed the lowest 0-fold/4-fold ratio, followed by the other two non-hybrid apomictic species, *C. cathayensis* and *C. dabieshanensis*, while the sexually reproducing *C. tonkinensis* had the highest ratio (all $P < 2.2 \times 10^{-16}$; Fig. 4c).

We further quantified mutation load in four functional categories, including synonymous (SYN), tolerated (TOL), deleterious (DEL), and loss-of-function (LOF) mutations, based on SNP data from the same 195 individuals and two outgroups (*C. illinoinensis* and *C. kweichowensis*). For this analysis, we used SIFT4G predictions based on *C. hunanensis* haplotype E reference genome (derived from apomictic lineages) as the primary reference. Consistent with its hybrid origin, *C. hunanensis* showed markedly higher heterozygosity and lower homozygosity across the four mutation types than the other three species (all $P < 2.2 \times 10^{-16}$; Fig. 4d and Supplementary Fig. 27). Accordingly, this corresponds to a lower realized mutation load, which reflects the contribution of homozygous variants, whereas the excess heterozygous variants mainly contribute to the masked load. Some differences were also observed between its TD and YL morphotypes (Supplementary Fig. 28). Excluding the hybrid, *C. tonkinensis* displayed significantly lower heterozygosity and higher derived homozygosity than the two non-hybrid apomictic species (all $P < 2.2 \times 10^{-16}$; Fig. 4d). Specifically, derived homozygous variants accounted for 98.13% of LOF and 97.58% of DEL mutations in *C. tonkinensis*, compared to 95.31% and 92.75% in *C. cathayensis*, and 93.64% and 91.20% in *C.*

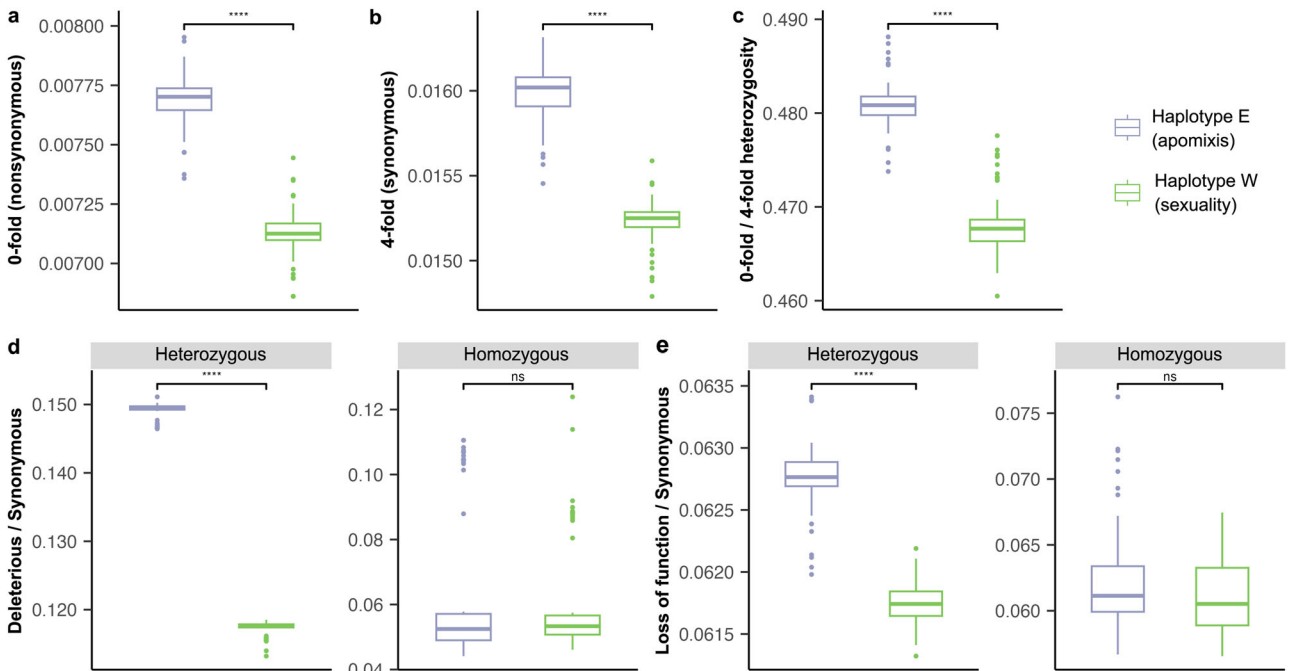

**Fig. 5 | Comparisons in inherited genetic load between the two haplotype genomes of the hybrid apomictic species *Carya hunanensis*.** Nucleotide diversity at nonsynonymous (0-fold degenerate) sites (**a**) and synonymous (4-fold degenerate) sites (**b**). **c** Ratio of heterozygosity at 0-fold sites relative to 4-fold sites. Ratios of derived deleterious (DEL) variants (**d**) and loss-of-function (LOF) variants (**e**) to synonymous variants in heterozygous and homozygous regions per individual. The two haplotype genomes represent different parental origins: haplotype E was inherited from the apomictic parents (*C. cathayensis* and *C. dabieshanensis*), while haplotype W was inherited from the sexual parent (*C. tonkinensis*). All analyses are based on 57 adult trees of *C. hunanensis*. Box plots summarize the distribution across individuals: the center line indicates the median, the box bounds correspond to the 25th and 75th percentiles, and whiskers extend to 1.5× the interquartile range. Statistical significance between haplotypes was assessed using two-sided Welch's t-tests without multiple-testing correction. Asterisks indicate significance levels (ns, not significant; *$P < 0.05$; **$P < 0.01$; ***$P < 0.001$). Source data are provided as a Source Data file.

*dabieshanensis*, respectively, calculated as homozygous-derived counted twice over the total derived, with the latter defined as homozygous-derived counted twice plus heterozygous-derived counted once. Notably, *C. cathayensis* carried more homozygous-derived and fewer heterozygous deleterious variants than *C. dabieshanensis* ($P < 2.2 \times 10^{-16}$ to $P = 2.4 \times 10^{-5}$; Fig. 4d). Gene Ontology (GO) enrichment analysis revealed that homozygous DEL and LOF mutations in the three apomictic species were enriched in multiple biological processes, including functions related to pollen recognition and reproductive development such as embryo, seed, and fruit development (Supplementary Data 4). Analyses based on Grantham scores[54] and Haplotype W (derived from sexual lineages) produced concordant overall patterns, confirming the robustness of these results (Methods; Supplementary Figs. 29–30).

### Inheritance of deleterious mutations in the hybrid apomict *Carya hunanensis* from sexual and apomictic parental lineages

To further explore the distribution of deleterious mutations inherited from sexual and apomictic parents, we compared the two haplotype-resolved genomes of the apomictic hybrid *C. hunanensis*: haplotype E, derived from the apomictic parents *C. cathayensis* and *C. dabieshanensis*, and haplotype W, derived from the sexual parent *C. tonkinensis* (Figs. 1–2 and Supplementary Data 3). Across 57 adult individuals of *C. hunanensis*, individual-level heterozygosity at 0-fold and 4-fold degenerate sites, as well as the 0-fold/4-fold ratio, were all significantly higher in haplotype E than in haplotype W (all $P < 2.2 \times 10^{-16}$; Fig. 5a–c), indicating greater retention of variation in the apomictic-derived genome. Consistently, the ratios of deleterious to synonymous variants (DEL/SYN and LOF/SYN) at heterozygous sites were also significantly elevated in haplotype E (all $P < 2.2 \times 10^{-16}$; Fig. 5d). In contrast, no significant differences were observed at

homozygous sites between the two haplotypes ($P = 0.58$ and 0.054; Fig. 5e).

### Mutation load in LOH regions reveals genomic costs of occasional recombination events in apomictic hickory species

Loss of heterozygosity (LOH) can unmask recessive deleterious mutations, thereby reducing individual fitness and potentially driving selection against recombination in asexual lineages[55,56]. To assess these genomic consequences, we compared deleterious mutation loads between LOH and non-LOH regions (heterozygosity cutoff of 0.001) in adults of the non-hybrid apomictic *C. cathayensis* and *C. dabieshanensis*, both exhibiting extensive genome-wide LOH (Supplementary Figs. 17–18, 21–22). On average, *C. cathayensis* exhibited more LOH blocks than *C. dabieshanensis* (1973 vs. 1805 with $P = 1.1 \times 10^{-6}$; Fig. 6a), consistent with its lower genome-wide heterozygosity. In both species, heterozygous DEL and LOF mutations were significantly reduced in LOH regions, typically with 0–6 sites. Notably, *C. cathayensis* harbored more heterozygous deleterious mutations than *C. dabieshanensis* (average 1.60 vs. 0.77 for DEL with $P = 2.3 \times 10^{-7}$; 3.36 vs. 0.60 for LOF with $P < 2.2 \times 10^{-16}$; Fig. 6b), reflecting differences in mutation accumulation. At homozygous sites, the ratio of deleterious to synonymous mutations (DEL/SYN) was significantly lower in LOH regions compared to non-LOH regions in both species (*C. cathayensis*: $P = 1.4 \times 10^{-5}$ and *C. dabieshanensis*: $P = 2.4 \times 10^{-5}$; Fig. 6c). However, homozygous LOF/SYN ratio showed divergent patterns: *C. cathayensis* had significantly higher ratios in LOH regions ($P = 3.6 \times 10^{-9}$), while *C. dabieshanensis* had lower ratios ($P = 2.2 \times 10^{-7}$; Fig. 6d), suggesting species-specific effects of recombination or its frequency.

To assess the genetic cost of recombination in apomictic hickories, we examined genome-wide mutation loads in 180 mature embryos of *C. cathayensis*, *C. dabieshanensis*, and the YL morphotype

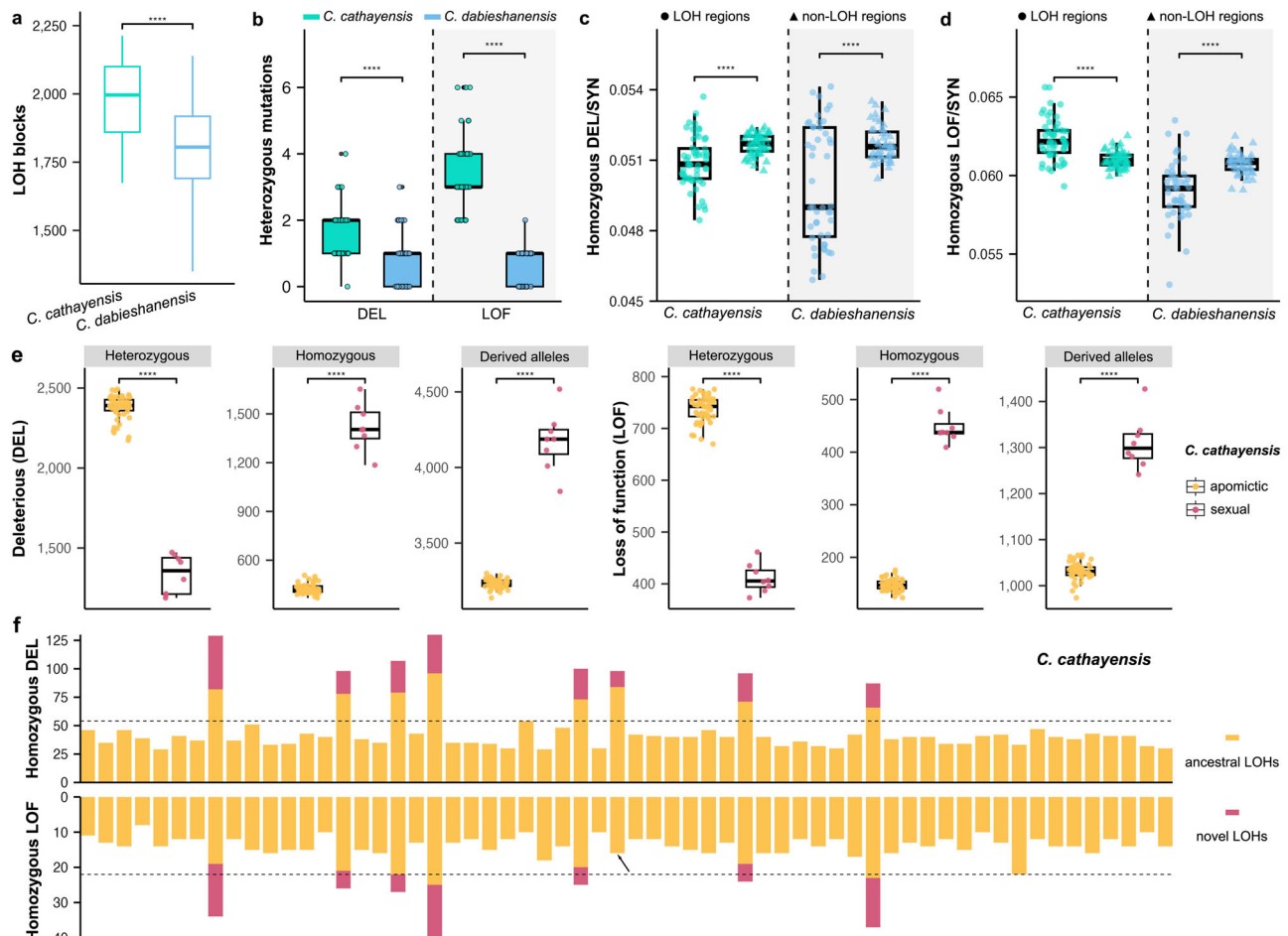

**Fig. 6 | Deleterious mutation accumulation associated with loss of heterozygosity (LOH) in apomictic hickories. a** Comparison of LOH blocks across the whole genome in adult individuals of the two non-hybrid apomictic species *Carya cathayensis* and *C. dabieshanensis*. **b** Differences in the accumulation of heterozygous deleterious (DEL) and loss-of-function (LOF) mutations within LOH regions between *C. cathayensis* and *C. dabieshanensis*. Comparison of the ratios of homozygous DEL (**c**) and homozygous LOF (**d**) mutations to synonymous (SYN) mutations in LOH versus non-LOH regions in *C. cathayensis* and *C. dabieshanensis*. **a–d** analyses are based on 55 adult individuals of *Carya cathayensis* and 47 adult individuals of *C. dabieshanensis*. **e** Comparison of deleterious mutation loads between 52 apomictic and 8 sexual mature embryos of *C. cathayensis*. **f** Numbers of homozygous DEL and LOF mutations in ancestral LOH regions (orange bars; shared across all 60 embryos) versus novel LOH regions (red

bars; arising via recombination in the 8 sexual embryos of *C. cathayensis*). The dashed line indicates the maximum number of homozygous deleterious mutations observed in apomictic embryos, and the arrow highlights the absence of LOF mutation accumulation in novel LOH regions of sexual embryos. LOH and non-LOH regions were defined using 50-kb sliding windows. For (**a–e**), box plots summarize the distribution across individuals or embryos: the center line indicates the median, the box bounds correspond to the 25th and 75th percentiles, and whiskers extend to 1.5× the inter-quartile range. Statistical significance was assessed using two-sided Welch's t-tests without multiple-testing correction. Asterisks indicate significance levels (ns, not significant; *$P < 0.05$; **$P < 0.01$; ***$P < 0.001$). Source data are provided as a Source Data file.

of *C. hunanensis*. Most embryos were genetically identical to their maternal plants, consistent with clonal inheritance, but a small subset showed genome-wide LOH patterns indicative of residual sexual reproduction (Fig. 3d–f). These sexual embryos displayed markedly reduced genome-wide heterozygosity (Supplementary Fig. 31) and increased homozygosity at deleterious mutation sites, including both DEL and LOF variants ($P = 1.5 \times 10^{-8}$ to $2.8 \times 10^{-6}$; Fig. 6e and Supplementary Figs. 32–33).

We next focused on LOH regions to test whether mutation load is further amplified within these vulnerable genomic segments. We compared the number of homozygous DEL and LOF mutations within ancestral LOH regions, which are shared by both sexual embryos and their apomictic siblings, and in novel LOH regions that occur only in sexually derived individuals. In most cases, sexual embryos carried substantially more homozygous deleterious variants within both ancestral and novel LOH regions (Fig. 6f and Supplementary Figs. 34–35). On average, sexual individuals of *C. cathayensis* exhibited 2.2–3.3-fold increases in homozygous DEL and 1.1–3.0-fold increases in

LOF burden relative to apomictic individuals within LOH regions (Fig. 6f). The corresponding increases in *C. dabieshanensis* were 1.7–2.5-fold (DEL) and 1.2–2.1-fold (LOF) (Supplementary Fig. 34). The most pronounced effects were observed in the hybrid taxon *C. hunanensis*, with 3.6–45.6-fold (DEL) and 4.3–84.1-fold (LOF) increases in homozygous mutations within LOH regions (Supplementary Fig. 35). Nonetheless, inter-individual variability was observed: in one sexual embryo each of *C. cathayensis* and *C. dabieshanensis*, no novel LOF mutations were detected, and in one *C. hunanensis* embryo, the deleterious burden was only slightly elevated compared to its clonal counterparts (Fig. 6f and Supplementary Figs. 34–35).

## Discussion

In this study, our analysis of large-scale genomic data confirms that *C. hunanensis* is an apomictic hybrid, likely originating from infrequent hybridization events between sexual reproducing *C. tonkinensis* and an apomictic lineage comprising *C. cathayensis* and *C. dabieshanensis*. This inference is supported by prior phylogenomic studies[47], field

observations of intermediate morphology, and experimental evidence from nucellar embryony, controlled pollination, and polyembryony occurrence[42–44,46]. Traditional approaches for detecting apomixis, including trait-based assessments for sporophytic types, as previously reported in *C. cathayensis*[42–44,46], and flow cytometric seed screening (FCSS) for gametophytic types[57], are informative but often labor-intensive, technically demanding, and constrained by species-specific phenology. We identify three population-level genomic features that strongly indicate clonal reproduction in *C. cathayensis*, *C. dabieshanensis*, and *C. hunanensis* but not in sexually reproducing *C. tonkinensis*. These features are genome-wide duplicated kinship structure, atypical bimodal/multimodal site frequency spectra, and uniform loss of heterozygosity. Together, these genomic signatures provide a robust and scalable framework for detecting clonal reproduction in natural populations, offering a valuable alternative to traditional phenotypic methods, particularly in non-model organisms, where reproductive modes may be cryptic or lineage-specific.

Unlike most hybrid species that arise from sympatric or parapatric parental taxa[58,59], *C. hunanensis* and its putative parental species are now strictly allopatric, each occupying a narrowly restricted geographic range (Fig. 2a). This distribution pattern suggests a scenario of historical hybridization followed by subsequent geographic isolation, potentially influenced by environmental changes. Supporting evidence includes herbarium records documenting past range extensions[60], and species distribution models that reconstruct potential contact zones[48,61]. At present, the pervasive clonal structure observed across extensively sampled populations indicates that asexual reproduction has become predominant in all three apomictic hickory species. This reliance on apomixis may confer adaptive advantages under geographic isolation and ecological constraints. Despite their underlying genetic diversity, as reflected by eight distinct genetic groups in *C. dabieshanensis* and four in *C. hunanensis* (Fig. 3b, c and Supplementary Figs. 18–19, 22), these apomictic lineages remain poorly characterized in terms of morphology and ecology. To date, only the TD and YL morphotypes have been identified in *C. hunanensis*. Given their economic potential as nut crops, coupled with their narrow distributions[39,40,62], clonal dominance, and cryptic diversity, these species warrant urgent attention through targeted field surveys, germplasm collection, and conservation planning.

While asexual reproduction can offer short-term ecological advantages[26], it also poses potential long-term genetic risks. The absence of recombination makes clonal lineages theoretically susceptible to Muller's ratchet, whereby reduced selection efficacy allows deleterious mutations to accumulate irreversibly[31,33,63]. Empirical studies in plants such as *Oenothera*[30] and *Boechera*[22] have confirmed this risk, showing elevated mutational loads in asexual lineages compared to their sexual relatives. However, our investigation of the apomictic hybrid hickory complex presents a markedly different case.

In contrast to their sexually reproducing relative *C. tonkinensis*, the three apomictic species exhibit lower realized mutational burdens, with deleterious alleles largely retained in the heterozygous state, and maintain comparable or even enhanced selection efficacy, as reflected by reduced 0-fold/4-fold ratios. Among them, the hybrid apomict *C. hunanensis* is particularly notable, showing genomic patterns consistent with expectations under the hybrid buffering effect[64], whereby admixture between highly divergent parental genomes promotes fixed heterozygosity[65]. While our observations align with this theoretical framework, we note that direct functional validation (e.g., gene expression assays) will be required to confirm the underlying mechanism. Haplotype-resolved analyses further reveal an asymmetric distribution of genetic load in *C. hunanensis*, with the apomictic-derived haplotype carrying a significantly higher proportion of heterozygous deleterious variants. This imbalance may reflect the evolutionary consequences of long-term asexuality, where suppressed recombination and relaxed purifying selection permit the persistence

of recessive deleterious mutations[33]. Despite this, the comparable levels of homozygous deleterious variants between the two haplotypes suggest that purging of strongly deleterious mutations may still be similarly effective in both sexual and asexual lineages. In asexuals, this may occur because large homozygous tracts are occasionally generated and, if deleterious, eliminated in bulk, leaving the surviving regions largely purged. However, even rare recombination events can unmask these hidden variants and increase the efficacy of purifying selection in hybrid apomictic systems[66]. This mechanism is supported by our findings in *C. hunanensis*, where LOH regions identified in two sexually derived embryos exhibit a striking enrichment of homozygous deleterious mutations.

Beyond the effects of hybridization, the two non-hybrid apomictic lineages *C. cathayensis* and *C. dabieshanensis* also show signs of purging deleterious mutations, suggesting that apomixis does not necessarily impede genome integrity. Notably, these species retain residual meiotic recombination and mixing processes, likely through occasional sexual reproduction. Clear signature of recombination was identified in mature embryos of all three apomictic species, and normal pollen development in *C. cathayensis*[67] further supports the hypothesis that meiosis and recombination remain at least partially functional. Such residual sexuality may enable purifying selection by exposing deleterious alleles during the haploid gametophytic phase[7,33]. While some sexually derived mature embryos showed elevated mutation loads, newly formed LOH regions in two such embryos—one from *C. cathayensis* and one from *C. dabieshanensis*—contained no homozygous LOF mutations. This pattern is suggestive of selection acting at multiple stages, with some genotypes carrying highly deleterious alleles possibly purged early, whereas other sexual embryos still retained homozygous LOF mutations, indicating that selection may also act later in development or during seedling establishment. Additional support comes from clonal genotyping, which identified five and two novel clonal lineages among 47 *C. dabieshanensis* and 57 *C. hunanensis* individuals, respectively, which exhibit first- to third-degree genetic relatedness. The reproductive biology of *Carya* further facilitates this mechanism: as monoecious, dichogamous, wind-pollinated trees, they promote outcrossing[41]. With each adult tree produces thousands of fruits per year, even rare sexually derived progeny with advantageous genotypes may spread rapidly, giving rise to new clonal genotypes. Our genomic findings, together with previous results from the apomictic hexaploidy *Ranunculus auricomus* complex[37], suggest that residual recombination may be sufficient to halt Muller's ratchet[36].

Despite both being apomictic and closely related, *C. cathayensis* and *C. dabieshanensis* exhibit distinct genomic responses to loss of heterozygosity (LOH), reflecting lineage-specific patterns of genome maintenance. LOH can unmask recessive mutations, thereby facilitating the purging of mildly deleterious variants, but it may also unmask strongly deleterious alleles, increasing genetic load[68,69]. In both species, LOH regions show a consistent depletion of heterozygous deleterious mutations, suggesting that LOH can act as a selective filter during the transition to homozygosity. However, their broader genomic context differs substantially. In adult *C. cathayensis*, widespread LOH, low genome-wide heterozygosity, and the dominance of a single clonal lineage indicate suppressed recombination and reduced efficiency of purifying selection. This genomic background is further characterized by an elevated mutation burden, increased 0-fold/4-fold ratios, and the accumulation of homozygous loss-of-function (LOF) mutations within LOH regions, all of which point to the long-term costs of clonal propagation when recombination is suppressed or absent[33]. By contrast, adult *C. dabieshanensis* maintains higher genomic heterozygosity, lower LOH coverage, and greater clonal diversity, alongside a lower mutation burden, reduced 0-fold/4-fold ratios, and more effective deleterious mutation purging in LOH regions. These features are consistent with more frequent recombination or residual sexual

reproduction, as supported by the higher frequency of sexually derived embryos (11/60 vs. 8/60 in *C. cathayensis*). Such recombination may facilitate the emergence of genetically distinct adult individuals (5 clones vs. 1 in *C. cathayensis*), potentially contributing to the long-term maintenance of genomic stability. Collectively, these findings illustrate that while apomixis ensures reproductive assurance, its evolutionary persistence depends on the interplay between clonality, recombination, and selection.

Overall, we found that apomictic hickory lineages maintain relatively effective purifying selection and a lower realized genetic load compared to their sexual relatives, highlighting the mitigating roles of residual recombination, hybrid origin, and clonal diversity in counteracting the genetic risks of asexuality. While recombination-mediated LOH can expose deleterious mutations, it may also generate viable novel genotypes, as suggested by the absence of deleterious homozygous mutations in certain mature embryos and the presence of multiple genetically distinct clonal lineages. Together, these mechanisms help prevent the accumulation of harmful mutations, enhancing the long-term persistence and adaptive potential of apomictic lineages. These results challenge the long-held assumption that apomixis inevitably leads to genomic decay and extinction. However, full reversion to sexual reproduction would be very costly by exposing the accumulated sheltered load, trapping apomixis in an evolutionary dead end.

## Methods

### Genome assembly, annotation, and haplotype phasing

To identify ploidy levels of the presumed hybrid taxon of *C. hunanensis*, fluorescence in situ hybridization (FISH) analysis was performed on root tip meristem tissues of the TD morphotype (Supplementary Figs. 1–2). Fresh young leaves of the TD morphotype of *C. hunanensis* (Tongdao County, Hunan Province, China; 26°6′ 53.63″N, 109°41′39.61″E) were collected from an adult tree. High-quality genomic DNA was extracted using the CTAB method. Illumina short-read (350 bp) and PacBio long-read (20 kb) libraries were sequenced on the NovaSeq 6000 and PacBio Sequel II platforms, respectively. Hi-C libraries were prepared and sequenced on the NovaSeq 6000. The quality-filtered Illumina short reads were used to estimate genome size, heterozygosity, and the repeat content of the genome by analyzing 19-mer frequencies using Jellyfish v2.3[70] and Genomescope v2.0[71] (Supplementary Fig. 3). High-accuracy CCS reads were used to assemble the draft genome with hifiasm v0.16[72]. Chromosome-level assembly was then performed using Hi-C data, with contigs clustered, ordered, and oriented using LACHESIS[73]. Misplacements and orientation errors were manually corrected based on chromatin interaction patterns, resulting in an assembly anchored to 32 chromosomes (Supplementary Fig. 4 and Supplementary Table 1). Genome assembly completeness was assessed by mapping Illumina whole-genome sequencing reads using BWA v0.7.12[74] and by BUSCO v5.2.2[75] with the "embryophyta_odb10" database (Supplementary Fig. 5). Gene prediction was conducted through a combination of ab initio prediction, homology-based inference, and transcriptome data from RNA sequencing. Repetitive sequences, including transposable elements (TEs) and tandem repeats, were also identified using a combination of homology-based and ab initio approaches. See details of this section in Supplementary Methods.

The initial assembly obtained was twice the size of the anticipated genome size (Supplementary Fig. 3), suggesting that it contained two haplotypes. Since *C. hunanensis* is thought to be a hybrid between *C. tonkinensis* and the ancestor of *C. cathayensis* and *C. dabieshanensis*[47], we separated these haplotypes using a method based on individual genome coverage. Illumina resequencing reads from five representative individuals of *C. cathayensis*, *C. dabieshanensis*, the TD and YL morphotypes of *C. hunanensis*, and *C. tonkinensis* were aligned to the

16 pseudochromosome pairs. The highest coverage from *C. cathayensis* and *C. dabieshanensis* was used to define homologous chromosomes as ChrA, with regions of lower coverage designated as ChrB (opposite coverage patterns were observed in *C. tonkinensis*). To validate the accuracy of haplotype phasing, haplotype-specific repetitive DNA sequences were identified using SubPhaser[49] with default parameters. This analysis confirmed the haplotypes identified by the genome coverage-based method. Consequently, the *C. hunanensis* genome was divided into two haplotypes, referred to as *C. hunanensis* haplotype E (East) and *C. hunanensis* haplotype W (West) (Fig. 1), reflecting their inferred parental geographic origins.

### Sample collection for leaf and fruit tissues and genome sequencing

To explore genetic details associated with apomixis, we conducted whole-genome resequencing on leaf tissue from 131 individuals across four *Carya* species. This included 41 *C. cathayensis*, 26 *C. dabieshanensis*, 36 *C. hunanensis* (comprising both TD and YL morphotypes), and 28 *C. tonkinensis* (Supplementary Data 1). Samples were collected from the natural distribution ranges of these species, ensuring that all sampled trees were mature (over 50 years old) and located in their native habitats. Field identification was primarily based on the morphological characteristics of leaves, fruits, and trunks. Genomic DNA was sequenced on the Illumina NovaSeq 6000 platform, generating 150 bp paired-end reads with an average depth of 30× for leaf samples. In addition to the newly sequenced data, we integrated previously published resequencing data from Zhang, Ding[47] and Zhang, Yang[48], which included 14 individuals of *C. cathayensis*, 21 *C. dabieshanensis*, 21 *C. hunanensis* and 8 *C. tonkinensis*. This resulted in a total sample size of 195 individuals: 55 *C. cathayensis*, 47 *C. dabieshanensis*, 57 *C. hunanensis*, and 36 *C. tonkinensis* (Supplementary Data 1). Furthermore, to detect potential recombination events mediated by residual sexuality in putatively apomictic taxa, we collected 180 mature embryo samples from three trees per taxon, with approximately 20 fruits per tree, for *C. cathayensis*, *C. dabieshanensis*, and the YL morphotype of *C. hunanensis* (Supplementary Data 2). These fully developed embryos from the fruit samples were sequenced using the same platform and protocol as the leaf samples but with an average depth of 10×.

### Read mapping and variants calling

The raw reads from leaf and mature embryo samples were trimmed for adapters, bases with a quality score below 20, and any reads shorter than 50 bp using Trimmomatic v0.38[76]. The remaining high-quality paired-end reads were aligned to the *C. hunanensis* haplotype E reference genome using the BWA-MEM algorithm in BWA v0.7.12[74] with default parameters, unless otherwise specified for mapping to haplotype W (see relevant sections below). SAMtools v0.1.19[77] was employed to retain uniquely and properly paired reads, and to convert the Sequence Alignment Map (SAM) format to Binary Alignment Map (BAM) format. For both sample types, the SENTIEON DNAseq software package v202112.05[78] was used to remove duplicate reads, realign indels, and perform single nucleotide polymorphism (SNP) calling for each individual. Joint SNP calling was conducted only for the leaf samples.

For both leaf and embryo samples, SNPs were filtered following the criteria established by Zhang, Cao[58]. The filtering process involved: (1) retaining only biallelic sites; (2) excluding sites with a mapping depth below one-third or above twice the individual's average depth (with a minimum depth cutoff of 6× for leaf samples and 3× for embryo samples), while applying Q20 quality score thresholds and heterozygosity adjustments[79]; (3) retaining all SNPs regardless of missing rate by default, although in certain analyses, more stringent filters were applied, such as excluding SNPs with >20% missing data or removing all SNPs with any missing genotypes; (4) eliminating singleton variants;

(5) excluding SNPs located within coding regions or their 20-kb flanking sequences to ensure neutrality; and (6) thinning SNPs by retaining a single site within each 20-kb window. For specific downstream analyses, custom SNP filtering steps were further applied based on the requirements of the software tools used and computational constraints.

## Population structure and genetic differentiation analysis

To assess the genetic relatedness among individuals and taxa from four *Carya* species, Bayesian clustering was performed using ADMIXTURE[50] based on 4,828 pruned SNPs from 195 individuals (filtered using steps 1–6, excluding SNPs with >20% missing data). ADMIXTURE employs a maximum likelihood-based model to estimate ancestral components in unrelated individuals. The number of genetic clusters ($K$) was predefined from 1 to 8, and cross-validation (CV) error was used to determine the optimal $K$ value. Additionally, principal component analysis (PCA) was also conducted on the same SNP dataset using the R package SNPRelate v1.34.1[80] with default parameters. To quantify genetic divergence between species and within-species lineages (as inferred by ADMIXTURE and PCA), we estimated interspecific $F_{ST}$ and $d_{xy}$ in non-overlapping 20-kb windows using pixy[81]. Since pixy requires both polymorphic and monomorphic (invariant) sites for unbiased estimation of $\pi$ and $d_{xy}$, we regenerated a separate VCF dataset for all 195 individuals using the GATK tools CombineGVCFs and GenotypeGVCFs tools in GATK v.4.4.0.0[82] with the "-all-sites" option enabled. After applying filtering steps 1–3 while retaining sites with <20% missing data, this pipeline produced a VCF dataset with invariant sites used for the estimation of genetic variation in pixy[81].

## Testing for hybridization events

Hybridization events were detected using HyDe software[51], which employs Paterson's $D$ statistics based on ABBA-BABA site patterns. The parameters γ and 1-γ denote the inheritance probabilities from two parental sources. Generally, γ values near 0.5 indicate a recent or infrequent hybridization event, while values closer to 0 or 1 suggest an ancient hybridization event that has persisted in present-day species. The analysis was conducted in two phases using a SNP dataset of 23,395 sites from 195 individuals and the two aforementioned outgroups. Filtering steps 1–4 and 6 were applied, and sites with missing data were removed due to computational constraints. Initially, based on phylogenetic results from Zhang, Ding[47], *C. cathayensis* and *C. dabieshanensis* were treated as a species complex, *C. tonkinensis* remained unchanged, and *C. kweichowensis* and *C. illinoinensis* served as outgroups to focus on hybridization within *C. hunanensis*. Subsequently, we extended the analysis to the aforementioned four species, using the same outgroups, to explore reticulation among these lineages.

## Maternally inherited plastid genome analysis

The high-quality paired-end reads from the 195 individuals, along with one individual of outgroup *C. illinoinensis*, were mapped to their respective species' chloroplast genomes (http://cmb.bnu.edu.cn/juglans) using the BWA-MEM algorithm in BWA v. 0.7.12[74]. After generating the realigned BAM files as described above, the 'mpileup' and 'call' functions in BCFtools v.1.15[83], along with the 'vcf2fq' command from vcfutils.pl, were used to convert these BAM files into VCF format and derive consensus chloroplast genome sequences. From these sequences, 80 shared protein-coding genes were extracted across all individuals based on their corresponding chloroplast genome annotations. Phylogenetic trees were reconstructed from the concatenated alignments of the 80 chloroplast genes using IQ-TREE v. 2.1.3[84] with the '-m MFP' model and 1000 rapid bootstrap replicates. Additionally, haplotypes for all individuals were identified from the concatenated sequences using DnaSP v6[85], and a median-joining (MJ) haplotype network was constructed using PopART v. 1.7[86].

## Germination test for polyembryony identification

To examine the occurrence of the multi-seedling phenomenon, an indicator of polyembryony and a potential sign of sporophytic apomixis[44,45], a germination experiment was conducted with four *Carya* species, including the newly identified TD and YL morphotypes of *C. hunanensis* (Supplementary Fig. 1). Fresh mature fruits were collected from six randomly selected trees per taxon, with 20 fruits harvested from each tree, resulting in 120 fruits per taxon. The fruits were stored at 4 °C for 24 h, then soaked in water for 2–3 h at room temperature. Following this, the fruits were buried in peat soil at room temperature under a 1–2 cm soil layer. Germination was monitored every two days, recording the number of seeds germinated until no further germination occurred after 30 days. The total number of seeds exhibiting the multi-seedling phenomenon and its occurrence rate were calculated for each taxon.

## Kinship inference and loss of heterozygosity analysis

We utilized the KING v.2.1 software[87] to infer family relationships and assess inbreeding levels within each of the four hickory species. This software estimates kinship coefficients independently of sample composition or population structure, based on the difference between shared heterozygosity and shared homozygosity. For accurate kinship inference, it is recommended to use genome-wide SNP data without prior pruning or filtering of "good" SNPs that pass quality control[87]. To maximize mapping rates and SNP recovery, all adult individuals of *C. cathayensis*, *C. dabieshanensis*, and *C. hunanensis* were mapped to the *C. hunanensis* haplotype E reference genome, while all *C. tonkinensis* individuals were mapped to the *C. hunanensis* haplotype W reference genome (Supplementary Data 1). The initial filtering steps 1–3 were applied, retaining sites with missing data, resulting in datasets of 6,515,619 SNPs for *C. cathayensis*, 9,506,975 SNPs for *C. dabieshanensis*, 23,681,021 SNPs for *C. hunanensis*, and 20,859,321 SNPs for *C. tonkinensis*. Kinship coefficients were calculated using the '-kinship' option with default settings in KING. Relationship classifications were assigned based on the ranges specified in the KING user manual, where kinship coefficients greater than 0.354 correspond to duplicate/MZ twins, 1st-degree relationships range from 0.177 to 0.354, 2nd-degree relationships range from 0.0884 to 0.177, and 3rd-degree relationships range from 0.0442 to 0.0884.

For genome-wide heterozygosity analyses, heterozygosity was calculated in 50-kb sliding windows for each of 195 adults and 180 embryos, using gVCF files as input to a custom Perl script. All samples were mapped to the *C. hunanensis* haplotype E reference genome to provide a consistent analytical framework, allow direct comparisons across tissues and reproductive modes, and minimize biases associated with using different references. Within each window, heterozygosity was calculated as the number of heterozygous sites divided by the total number of non-missing sites. To minimize technical artifacts, sites were excluded if sequencing depth fell outside an individual-specific range defined as one-third to twice the average coverage, but with a minimum cutoff of 6× for adults ( ~ 30× coverage) and 3× for embryos ( ~ 10× coverage). Multiallelic and indel sites were removed, and heterozygous genotypes were further filtered by allele balance[79], requiring allelic ratios of 0.2–0.8 at higher depths ( ≥20× for adults, ≥6× for embryos) and 0.1–0.9 at moderate depths ( ≥10× for adults, ≥3× for embryos), otherwise a homozygous call was assigned. Finally, gVCF block coordinates were expanded per base to ensure alignment, and windows with >50% missing sites were reported as NA.

Regions of loss of heterozygosity (LOH) were defined as 50-kb windows with heterozygosity <0.001, a cutoff benchmarked against the sexual control *C. tonkinensis*, where only 2.01% of 345,979 windows fell below this value (Supplementary Figs. 15–16). This sexual baseline provides a conservative threshold, minimizing false positives (rare even under continuous meiotic recombination) and avoiding circularity that could result from defining cutoffs on apomictic genomes

enriched in homozygous tracts. After the stringent filtering described above, the rare windows with heterozygosity <0.001 observed in sexual likely reflect genuine regions of low diversity rather than technical artifacts. To confirm robustness, we repeated the analyses using a more stringent cutoff of 0.0005, which applied to only 0.67% of windows in the sexual control.

Following the rationale of Weir, Capewell[88] and Simion, Narayan[89], which examined LOH to investigate mitotic recombination in asexual reproduction, we adapted this approach to assess meiotic recombination in apomictic hickories. In principle, sexual reproduction is expected to generate variable LOH patterns among individuals through recurrent recombination, whereas clonal reproduction should yield highly consistent LOH profiles within lineages. To evaluate whether the LOH detected in apomictic species could instead reflect residual sex, we contrasted our data with expectations from alternative mechanisms: mitotic repair, which typically generates short and localized tracts unlikely to occur simultaneously across multiple chromosomes[52], and automixis, a modified meiosis predicted to produce genome-wide but structured LOH with positional biases[4,53] (schematic comparison in Supplementary Fig. 14). For the 195 adults, species- and lineage-level LOH profiles were compared, with lineage-shared large blocks spanning all 16 chromosomes interpreted as potential recombination events that had subsequently become clonally fixed. To further test whether LOH could arise from residual sexual reproduction, we analyzed 180 embryos (60 per apomictic species). In this framework, the occurrence of a minority of embryos with LOH profiles distinct from the predominant clonal background was interpreted as evidence of rare recombination events arising during sexual reproduction. The genomic distribution of LOH blocks for each individual was visualized using a custom R script.

## Folded site frequency spectrum and genetic diversity analysis

After identifying the reproductive modes and clonal lineages of the four hickory species, we assessed and compared their genomic variation. First, intraspecific nucleotide diversity ($\pi$) for adults from all four species (including the TD and YL morphotypes of *C. hunanensis*) was calculated using pixy[81], incorporating both invariant and variant sites within a 20-kb sliding window. Second, individual heterozygosity ($h$) was calculated as the number of polymorphic sites divided by the total length of the *C. hunanensis* haplotype E reference genome, based on a filtered SNP dataset comprising 39,934,221 sites from 195 individuals (filtering steps 1–3). Third, we calculated the folded site frequency spectrum (SFS) for each taxon using a Perl script from Zhang, Cao[58], based on 6,132,224 SNPs from the same set of individuals. SNPs with missing data were removed during filtering to minimize biases in allele frequency estimates and ensure accurate SFS reconstruction.

## Evaluation of genetic load across sexual and asexual reproductive types

We calculated two metrics to quantify and compare the selection efficacy and deleterious genetic load across species. First, we determined the genome-wide ratio of heterozygosity (heterozygotes/total genotypes) for 0 to 4-fold degenerate sites within coding regions, based on the *C. hunanensis* haplotype E reference genome annotation. We then calculated the ratio of 0-fold nonsynonymous to 4-fold synonymous degenerate sites for each individual. Second, we annotated the effects of SNP variants (56,274,446 SNPs from 195 individuals and two outgroups, filtered using steps 1–3) on protein-coding genes, classifying them as synonymous, missense, or loss-of-function (LOF) variants using SnpEff v.5.0[90]. LOF variants were defined as those involving a gain and/or loss of stop codons, as well as the loss of start codons. Missense SNPs were further classified as deleterious (score ≤ 0.05) or tolerated (score > 0.05) using SIFT4G program with custom databases built from the UniRef90 protein dataset[91]. To avoid reference bias in identifying derived alleles and deleterious variants, we

determined the polarity of variants only when the two outgroups, *C. illinoinensis* and *C. kweichowensis*, shared identical homozygous states. For each SNP position in the SNP dataset, we determined the ancestral and derived allelic states using Est-SFS software[92], applying a probability cutoff of 0.95 and assuming a Rate-6 substitution model. The ancestral alleles were assumed to be non-deleterious, while the derived alleles were considered potentially deleterious. We calculated the total number of derived alleles by counting heterozygous genotypes once and homozygous-derived genotypes twice. To verify robustness, we additionally estimated deleterious burden with Grantham scores[54], which quantify the degree of amino acid change, designating missense mutations with scores ≥150 as deleterious and <150 as tolerated[93,94]. We further repeated SIFT4G predictions on the same 197 individuals mapped to the alternative haplotype W reference genome (58,940,457 SNPs after identical filtering). Both approaches yielded results consistent with the SIFT4G predictions based on the Haplotype E reference; therefore, we retained the Haplotype E–based SIFT4G results in the main text and used them for subsequent analyses. Gene ontology (GO) enrichment analysis for genes associated with DEL and LOF mutations in apomictic hickory species was performed using Tbtools[95] with default parameters.

## Comparison of inherited genetic loads between two genome haplotypes in *Carya hunanensis*

Building on our identification of hybrid diploid apomixis in *C. hunanensis*, we performed a comparative analysis of mutation burdens inherited from the two parental haplotypes. All 57 sampled adult *C. hunanensis* individuals were separately aligned to the haplotype E (derived from the apomictic parent) and haplotype W (derived from the sexual parent) reference genomes. After applying SNP filtering steps 1–3, we obtained 43,020,882 and 42,202,704 SNPs from 59 individuals (including two outgroups) for haplotype E and haplotype W, respectively. For each haplotype, we calculated the genome-wide nonsynonymous (0-fold) and synonymous (4-fold) site counts within coding regions and determined the nonsynonymous/synonymous (0-fold/4-fold) mutation ratio, following the same approach as previously described. Additionally, to better compare the difference in deleterious sites between the two haplotype genomes, we focused on the ratios of deleterious to synonymous mutations (DEL/SYN) and loss-of-function to synonymous mutations (LOF/SYN), considering both heterozygous and homozygous variants.

## Quantification of mutation loads in LOH regions of adult trees and mature embryos

Loss of heterozygosity (LOH) can reduce individual fitness by unmasking recessive deleterious mutations in diploid genomes. This mechanism has been proposed as a contributing factor to the rarity of asexual organisms in nature and may drive selection for reduced recombination rates in asexual lineages[55,56]. Given the extremely low distribution of LOH regions in the hybrid apomictic species *C. hunanensis*, our analysis focused on *C. cathayensis* and *C. dabieshanensis* to investigate patterns of deleterious variation associated with LOH. Using whole-genome resequencing data from adult individuals, we applied a 50-kb sliding window with a heterozygosity cutoff of 0.001 applied to classify genomic regions into LOH and non-LOH categories. To quantify the deleterious mutation burden, we analyzed homozygous and heterozygous sites separately. For homozygous sites, we counted the number of predicted deleterious (DEL), loss-of-function (LOF), and synonymous (SYN) mutations per window, and calculated mutation load as the ratios DEL/SYN and LOF/SYN for each region type. For heterozygous sites, because heterozygous variants were frequently absent in many windows, which precluded reliable ratio-based normalization, we quantified mutation burden by directly comparing the raw counts of heterozygous DEL and LOF mutations between LOH and non-LOH regions.

To assess the extent of new deleterious mutation accumulation in sexual embryos relative to their maternal plants, we further analyzed SNP data from 180 mature embryos (60 per species) across the three apomictic species. Apomictic embryos were treated as genetic equivalents of their maternal trees, as their genotypes are expected to be identical due to the absence of meiosis and fertilization. Accordingly, we generated 26,760,972, 26,722,247, and 34,948,656 SNPs (filter steps 1–3) from 60 individuals and two outgroup samples for *C. cathayensis*, *C. dabieshanensis*, and the YL morphotype of *C. hunanensis*, respectively. First, we compared the overall mutation burden between apomictic and sexual embryos to evaluate the potential genomic costs associated with sexual reproduction. The sexual embryos were identified in the above-mentioned loss of heterozygosity analysis, and their genetic load was assessed using the same methods as applied to adult individuals. Considering sample size imbalance in the YL morphotype of *C. hunanensis* (57 apomictic vs. 3 sexual embryos), we used the Wilcoxon rank-sum test for this comparison, whereas Welch's t-test was applied to adult comparisons (as described above) as well as to the other embryo contrasts (*C. cathayensis*: 52 vs. 8; *C. dabieshanensis*: 49 vs. 11). In addition, since LOH typically exposes recessive deleterious mutations to selection, we further classified LOH regions into two categories: (i) ancestral LOH, defined as regions shared by all embryos within the apomictic species, and (ii) novel LOH, referring to regions newly formed through recombination in sexual embryos. Within each LOH type, we quantified the number of homozygous deleterious (DEL) and loss-of-function (LOF) mutations. We hypothesized that sexual embryos exhibiting no significant increase in deleterious mutations within new LOH regions may represent novel genotypes with a higher likelihood of being retained by selection.

### Reporting summary

Further information on research design is available in the Nature Portfolio Reporting Summary linked to this article.

## Data availability

The raw sequence data reported in this paper have been deposited in the National Genomics Data Center (NGDC; https://ngdc.cncb.ac.cn) under the accession number PRJCA033579 and in GenBank under accession number PRJNA356989. The final genome assembly and genome annotation files are available at http://cmb.bnu.edu.cn/juglans. All other data supporting the findings of this study are present in the paper and/or its Supplementary Materials. Source data are provided with this paper.

## Code availability

The custom scripts used for the primary loss of heterozygosity (LOH) analyses are available at https://github.com/Hickory01/Apomixis-in-Hickory-Species.git.

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

## Acknowledgements

This work was supported by the National Natural Science Foundation of China (32400311 to W.P.Z.), the "111" Program of Introducing Talents of Discipline to Universities (B13008 to D.Y.Z.), the Fundamental Research Funds for the Central Universities (to D.Y.Z.), China Postdoctoral Science Foundation (2023M733549 to W.P.Z.) and the Chinese Academy of Sciences (CAS) Scholarship Program (to W.P.Z.). We thank Dr. Shou-Jie Li (South China Botanical Garden, Chinese Academy of Sciences), Dr. Yu Cao, Dr. Yang Yang and Dr. Rui-Min Yu (Beijing Normal University), Shi-Bin Jiao, You-Liang Xiang (Guizhou Normal University), Zhi-Quan Liu (Hangzhou Normal University), Kai-Bing Liu, and Zhi-Xiang Chen for their invaluable assistance with sample collection. We also thank Dr. Nan Wang (Agricultural Genomics Institute at Shenzhen, Chinese Academy of Agricultural Sciences) and Dr. Huiqin Yi (Stockholm University) for their insightful comments on the analysis.

## Author contributions

W.N.B., D.Y.Z., M.L., and M.K. conceived and supervised the project. W.P.Z. collected the materials and performed the main analyses. S.G. provided constructive suggestions regarding the loss of heterozygosity (LOH) analysis. X.X.P. conducted the LOH detection analysis. M.K. provided guidance on the analytical workflow for genetic load analysis. W.P.Z. and W.N.B. drafted the manuscript. W.N.B., M.L., S.G., and D.Y.Z. revised and proofread the manuscript. All authors approved the final version.

## Funding

## Competing interests

The authors declare no competing interests.
