## [Transparent Peer Review file · Nature Communications]

Genomic Consequences of Residual Recombination in a Hybrid Apomictic Hickory Complex

Corresponding Author: Dr Wei-Ning Bai

Version 0:

Reviewer comments:

Reviewer #1

(Remarks to the Author)

This manuscript investigates the genomic consequences of residual recombination in a hybrid apomictic hickory complex, examining clonal maintenance, mutation load, and the role of loss of heterozygosity (LOH). The study employs a combination of high-quality genome assembly, population genomic analyses of adult individuals, and embryo sequencing to address fundamental questions regarding apomixis and its long-term genetic implications. While the study presents a technically sound genomic analysis of *Carya apomixis*, it fails to meet the novelty threshold required for Nature Communications. The core findings represent incremental advances over prior work, and methodological limitations undermine key claims. I recommend rejection without resubmission for the reasons detailed below.

1. Insufficient Novelty and Conceptual Advance. The central premise—that residual sexuality mitigates mutation accumulation in apomicts—is well-established across multiple systems. For instance, studies in polyploid apomicts like *Ranunculus auricomus* (Pellino et al., 2013, *Mol. Ecol.*) demonstrated purging via LOH; research in diploid systems such as *Boehera* spp. (Lovell et al., 2017, *PLoS Genet.*) quantified reduced mutation loads in apomicts through heterozygosity masking; and work on hybrid taxa in *Citrus* (Wang et al., 2017, *Nat. Genet.*) linked heterosis to deleterious mutation buffering. This study applies these established concepts to a new taxonomic group (*Carya*) without proposing novel evolutionary mechanisms, challenging existing paradigms, or providing cross-system insights that redefine apomixis theory.

2. Methodological concerns undermine key conclusions. (a) LOH Definition and Validation: The classification of LOH as 50-kb windows with heterozygosity <0.001 is arbitrary and lacks justification. This threshold is not compared to sexual controls (e.g., *C. tonkinensis*) or validated against independent methods (e.g., allele frequency spectra), raising doubts about whether reported LOH regions truly reflect recombination or technical artifacts. (b) Sample Size Limitations: The inference of "residual sexual reproduction" relies on extremely small numbers of embryos with LOH (8 in *C. cathayensis*, 11 in *C. dabieshanensis*, 3 in *C. hunanensis*). This paucity of data makes it impossible to robustly quantify recombination rates or assess its evolutionary significance, weakening claims about "costs and benefits of residual sex." (c) Mutation Load Calculation: The use of SIFT scores to classify deleterious variants without accounting for lineage-specific functional constraints (e.g., divergence between hickory haplotypes) may misclassify neutral or adaptive mutations as deleterious, skewing comparisons between apomictic and sexual lineages.

3. Inconsistent Interpretation of Results. (a) It claims apomicts have "lower realized mutation loads" but acknowledges they harbor "more heterozygous deleterious variants." This tension is attributed to "hybrid buffering" in *C. hunanensis*, but no mechanistic evidence (e.g., gene expression data showing masked recessive effects) supports this assertion. (b) LOH is proposed to "expose deleterious mutations to selection," yet sexual embryos with LOH show higher homozygous deleterious loads. The suggestion that "selection eliminates harmful alleles early" is speculative and unsupported by data on embryo viability or fitness.

Reviewer #2

(Remarks to the Author)

The manuscript uses advanced kinship analysis of *Carya* species with dense SNP data and the KING algorithm to study clonality and possible apomixis. The authors apply the KING algorithm, which allows them to analyze unfiltered SNP datasets and look for duplicated (clonal), first, second, and third-degree kin relationships. The study claims that there is a lot

of clonality, sometimes almost complete, within these species.

By adding resequencing data from adult tissues and sequencing 180 embryos, the authors try to answer old questions about hybridization, persistent lineages, and the effect of apomictic versus sexual reproduction in this complicated hickory group. The study is relevant, the methodology is solid, and the evolutionary and ecological questions are important. Still, some points in the manuscript need to be explained better to make the conclusions clearer and reliable. Studying clonality on a genome-wide level in woody apomictic plants is new and important, including for the ecology and agriculture of hickory. The use of site frequency spectra (SFS) and loss-of-heterozygosity (LOH) analysis makes the argument for clonality even stronger and gives different views on the genetic structure.

The analysis shows clonality at the population level very well. Including 180 mature embryos from three apomictic species is a very valuable part of the study. The authors say that rare loss-of-heterozygosity (LOH) events in embryos are probably caused by leftover recombination from occasional sexual reproduction, not from mitotic mistakes. This explanation is reasonable and interesting, and it fits with the finding that there is no genome-wide LOH.

However, it would be helpful to better explain the difference between LOH caused by mitosis and LOH caused by automixis. For example, this could be done by showing allele imbalance of LOH patterns on chromosomes. My main doubts comes from the fact that *Carya* is a sexually outcrossing (allogamous) species. Generally, in diploid species, the level of heterozygosity is roughly halved every generation when selfing occurs. In contrast, cross-pollination keeps or even increases heterozygosity in the population, preserving genetic variation.

In outcrossers, selfing causes higher LOH with a fast increase of homozygosity and a decrease of heterozygosity. Cross-pollination results in less LOH and keeps heterozygosity and genetic diversity high. This is why allogamous species that do repeated selfing experience a quick loss of heterozygosity and often suffer inbreeding depression.

My personal idea to explain the results (embryos from sexual reproduction showing LOH), together with the authors' conclusion, is that in the crosses, both pollen and egg come from two plants of the same clone. But this should be studied more carefully. Still, seeing such LOH in only one generation (embryos from a mother plant) with crossing is unexpected. The authors should maybe sequence three biological replicates only for the samples that they think come from sexual reproduction, just to confirm that the LOH is consistent. I also suggest explaining this part more clearly. This is important because the embryo genotypes, especially related to LOH and recombination, are a large part of the study's analysis and interpretation. So, this section should be better described better in the main methods or detailed more clearly in the Supplementary Information.

Also, I suggest adding more discussion about other possible reasons, like mitotic recombination or technical problems.

In conclusion, this paper gives valuable information about apomixis, clonality, and how hybrid species in long-living trees stay stable through time. It just needs some data/revisions, especially to explain the embryo part better.

Version 1:

Reviewer comments:

Reviewer #1

(Remarks to the Author)

This study presents the first genome-wide evidence of recombination-mediated loss of heterozygosity (LOH) in a diploid woody apomictic complex (*Carya*), demonstrating how rare sexual events unmask deleterious alleles while contributing to genotypic diversity. The authors show that apomicts maintain lower realized mutation loads than sexual relatives, despite higher heterozygosity for deleterious variants, suggesting hybrid buffering and purging. Methodological concerns about LOH detection and mutation load estimation were robustly addressed through sexual control benchmarking, sensitivity analyses, and orthogonal validation. While the number of recombinant embryos is limited, their detection across three species confirms residual sexuality. The study fills a notable gap in apomixis research, which has largely focused on polyploid herbs. The manuscript is now substantially strengthened through revisions and supplementary analyses, and I support its acceptance pending minor textual clarifications.

Reviewer #2

(Remarks to the Author)

Authors have answered to most of my suggestions and the manuscript is clearer now almost all my comments were addressed.

RESPONSE TO REVIEWERS' COMMENTS

Reviewer #1 (Remarks to the Author):

This manuscript investigates the genomic consequences of residual recombination in a hybrid apomictic hickory complex, examining clonal maintenance, mutation load, and the role of loss of heterozygosity (LOH). The study employs a combination of high-quality genome assembly, population genomic analyses of adult individuals, and embryo sequencing to address fundamental questions regarding apomixis and its long-term genetic implications. While the study presents a technically sound genomic analysis of *Carya* apomixis, it fails to meet the novelty threshold required for Nature Communications. The core findings represent incremental advances over prior work, and methodological limitations undermine key claims. I recommend rejection without resubmission for the reasons detailed below.

Answer: We thank the reviewer for acknowledging that our study presents a technically sound genomic analysis of *Carya* apomixis. However, we respectfully disagree with the assessment that the work lacks sufficient novelty. As outlined in the detailed point-by-point responses below, our analyses provide the first genome-wide evidence of recombination-mediated LOH in a diploid woody apomictic complex and demonstrate its consequences for deleterious variation. We believe these findings have both evolutionary and applied significance. Methodological issues and questions regarding interpretation of results have also been addressed through clarifications and supplementary analyses, with corresponding revisions in the manuscript.

1. Insufficient Novelty and Conceptual Advance. The central premise—that residual sexuality mitigates mutation accumulation in apomicts—is well-established across multiple systems. For instance, studies in polyploid apomicts like *Ranunculus auricomus* (Pellino et al., 2013, *Mol. Ecol.*) demonstrated purging via LOH; research in diploid systems such as *Boechera* spp. (Lovell et al., 2017, *PLoS Genet.*) quantified reduced mutation loads in apomicts through heterozygosity masking; and work on hybrid taxa in *Citrus* (Wang et al., 2017, *Nat. Genet.*) linked heterosis to deleterious mutation buffering. This study applies these established concepts to a new taxonomic group (*Carya*) without proposing novel evolutionary mechanisms, challenging existing paradigms, or providing cross-system insights that redefine apomixis theory.

Answer: We note that the reviewer's interpretation of the three cited studies does not reflect the conclusions of the original articles, all of which were already discussed in our manuscript (Introduction, lines 62 and 72–75; Discussion, lines 373–376).

1. **Pellino et al. 2013 (*Mol. Ecol.*) did not demonstrate “purging via LOH.”**

The study analyzed **RNA-seq** data from the *Ranunculus auricomus* complex (2 diploid and 1 tetraploid sexual species, plus 2 apomictic hexaploid lineages) and therefore could not assess LOH from transcriptomic SNPs. More importantly,

their main conclusion was that **there is no genome-wide difference in deleterious mutation accumulation between apomictic and sexual lineages**, based on dN/dS comparisons of 1,231 genes from five representative individuals.

2. **Lovell et al. 2017 (*PLoS Genet.*) reports the opposite of the reviewer’s claim.** In **diploid/diploid-hybrid *Boechea***, Lovell et al. (2017) found that **apomicts harbor more derived mutations than sympatric sexuals**, and at phylogenetically constrained sites the ancestral (conserved) allele is more likely retained in sexuals—consistent with stronger purifying selection in sexual lineages. Thus, the paper supports **greater** (not reduced) mutation load in apomicts. Notably, the study did not explicitly control for **facultative apomixis** (variable proportions of sexual/asexual seeds were present in 11 of 13 apomictic samples, as shown in their Table S1), which can complicate inference about purely asexual dynamics.
3. **Wang et al. 2017 (*Nat. Genet.*) does not analyze deleterious load or “heterosis buffering.”**

The paper focuses on the genetic basis of **polyembryony in citrus** using genome and transcriptome data. Although some population-genomic contrasts among primitive, wild, and cultivated groups of genetic diversity were presented, the study **did not** quantify deleterious mutation burden nor link heterosis to buffering of deleterious variants.

Therefore, as we outlined in the manuscript, the broader empirical picture remains unclear. The notion that occasional or residual sex may reduce mutation accumulation has been proposed in theoretical models (e.g., Green & Noakes 1995, *J. Theor. Biol.*). However, empirical support from apomictic systems remains limited and not well-established. The only case study to date, in the *R. auricomus* complex (Hodač et al. 2019, *BMC Evol. Biol.*), genotyped three F_1 progeny arrays using six microsatellite markers and combined these data with mathematical modeling in a polyploid apomictic context—without genome-wide validation. As highlighted by Hörandl et al. (2020, *Crit. Rev. Plant Sci.*, “Genome Evolution of Asexual Organisms and the Paradox of Sex in Eukaryotes”), the interplay between facultative sexuality, selfing, and apomixis introduces further complexity into mutation dynamics in asexual plants—an area that remains poorly understood, largely due to the scarcity of genomic resources.

By contrast, our work provides—to our knowledge—the **first genome-scale empirical validation in apomicts** that **rare recombination-mediated LOH** occurs and shapes deleterious variation. We analyze a **diploid, woody, wild system (*Carya*)** that has been largely overlooked relative to previously studied apomictic systems, which are predominantly polyploid herbaceous plants or crops. Specifically, our study focuses on two non-hybrid apomicts (*C. cathayensis* and *C. dabieshanensis*), a sexual species (*C. tonkinensis*), and their hybrid derivative, the apomict *C. hunanensis*. Our dataset comprises a new phased reference genome for *C. hunanensis*, **195** resequenced adult trees and **180** mature embryos (60 per apomictic species). We validate reproductive modes and conduct **multi-level comparisons (sexual vs apomicts; haplotype contrasts within the hybrid apomict; apomictic parents compared to**

each other; apomictic vs sexual embryos). Critically, we introduce **genome-wide LOH analysis in apomicts** and show that embryos from apomictic mothers sometimes undergo **recombination-mediated LOH** that unmask recessive deleterious alleles (a genetic cost), while adult relatedness patterns indicate that occasional sex can generate novel genotypes that persist. Remarkably, despite harboring more heterozygous deleterious variants, apomictic adults showed **lower realized mutation loads** than their closely related sexual species. Thus, our study does not simply transpose an old concept onto a new taxon; rather, it provides the **first genome-wide evidence in a diploid woody apomictic complex** for how rare recombination influences mutation dynamics, with clear ecological and agricultural relevance.

2. Methodological concerns undermine key conclusions.

(a) LOH Definition and Validation: The classification of LOH as 50-kb windows with heterozygosity <0.001 is arbitrary and lacks justification. This threshold is not compared to sexual controls (e.g., *C. tonkinensis*) or validated against independent methods (e.g., allele frequency spectra), raising doubts about whether reported LOH regions truly reflect recombination or technical artifacts.

Answer: The issues regarding the LOH threshold, the use of sexual controls, and the possibility of technical artifacts were carefully considered in our analyses. To ensure clarity, we have now substantially expanded the description of these safeguards in the revised manuscript (Lines 628–654).

To define LOH, we used a cutoff of heterozygosity <0.001 , which was not arbitrary but benchmarked against the sexual control *C. tonkinensis*, as the reviewer rightly suggested. In this species, only 2.01% of ~346k 50-kb windows fell below this value, confirming that 0.001 lies in the extreme left tail of the sexual heterozygosity distribution (Fig. S15). A genome-wide heterozygosity plot (Fig. S16) further illustrates a generally stable background level punctuated by rare LOH segments where heterozygosity approaches zero. To test robustness, we also repeated the analyses using a more stringent threshold of heterozygosity <0.0005 (met by only 0.67% of windows in the sexual control) and obtained consistent LOH patterns (Figs. S20 and S23).

To minimize potential technical noise, heterozygosity was computed from per-base gVCF information with stringent filters: sites outside individual-specific depth ranges, indels/multiallelics, and heterozygotes failing allele-balance filters were excluded, while windows with $>50\%$ missing sites were reported as NA. These safeguards support the use of 0.001 as a conservative and biologically justified criterion for LOH detection. In the revised manuscript, the 0.001-based results are presented in the main text, with the 0.0005 cutoff included as supporting evidence (Lines 213–232).

Figure S15A. Distribution of window heterozygosity across sexual control *Carya tonkinensis*.

Histograms show heterozygosity values calculated in 50-kb windows for 36 resequenced adult individuals of *C. tonkinensis*. The dashed vertical line marks the predefined cutoff of 0.001 used to define loss of heterozygosity (LOH). Only 2.01% of 345,979 windows fall below this value, confirming that it represents the extreme left tail of the sexual heterozygosity distribution and provides a conservative threshold for LOH detection. More details can be found in the Methods.

(b) Sample Size Limitations: The inference of "residual sexual reproduction" relies on extremely small numbers of embryos with LOH (8 in *C. cathayensis*, 11 in *C. dabieshanensis*, 3 in *C. hunanensis*). This paucity of data makes it impossible to robustly quantify recombination rates or assess its evolutionary significance, weakening claims about "costs and benefits of residual sex."

Answer: Our study does not attempt to estimate recombination rates per se; rather, to establish the occurrence of recombination-mediated LOH in apomictic embryos and to evaluate its genetic consequences. Although the numbers of sexually derived embryos are modest (8, 11, and 3 out of 60 sampled per species), their detection across the three apomictic hickory species provides clear evidence of sexual reproduction-mediated recombination. In fact, a single embryo with LOH would have been sufficient to show that recombination occurs. In addition, since an individual apomictic adult hickory tree typically produces hundreds to thousands of fruits each year, the absolute number of fruits generated through sexual reproduction is still considerable. Therefore, once their occurrence is established, it allows us to reveal whole-genome patterns of LOH and mutation load in these 22 (3+8+11) sexually produced embryos. These patterns directly inform the genetic costs and potential benefits of residual sex, thereby making it meaningful to explore their evolutionary significance.

To avoid over-interpretation, we have revised the text to remove strong qualifiers such as *high* and adjusted the Abstract (Lines 27–31) to read: "Remarkably, rare

embryos from apomicts underwent recombination-mediated LOH, exposing deleterious mutations to selection. These findings reveal the genetic cost of residual sex, while also indicating its role in generating novel genotypes, supported by close relatedness among adult apomicts." We have also revised the relevant sections of the Results and Discussion for clarity.

(c) Mutation Load Calculation: The use of SIFT scores to classify deleterious variants without accounting for lineage-specific functional constraints (e.g., divergence between hickory haplotypes) may misclassify neutral or adaptive mutations as deleterious, skewing comparisons between apomictic and sexual lineages.

Answer: We thank the reviewer for this constructive point. We acknowledge that any single predictor may, in principle, misclassify some missense variants, particularly in cross-species contexts. Nevertheless, SIFT4G (Vaser et al. 2016, *Nat. Protoc.*) has been widely applied in comparable studies of non-model trees and other plants using a single reference genome—for example, in *Ostrya rehderiana* and congeners (Yang et al. 2018, *Nat. Commun.*) and in multiple *Populus* species (Liu et al. 2022, *Mol. Biol. Evol.*).

To further assess robustness, we evaluated our data using two complementary approaches: (i) applying an alternative predictor, Grantham scores (Grantham 1974, *Science*), to the same dataset mapped to the *C. hunanensis* Haplotype E genome (derived from apomictic lineages), and (ii) repeating SIFT4G analyses with the alternative haplotype, Haplotype W (derived from the sexual lineage). Both Grantham-based and haplotype-W-based predictions closely mirrored the patterns obtained with SIFT4G on Haplotype E (Figs. S4D, S29–S30), with the key conclusions unchanged: apomictic adults harbor more heterozygous deleterious variants but lower realized (homozygous) loads than the sexual relative *C. tonkinensis*. Therefore, we consider our results robust to both predictor choice and divergence between hickory haplotypes.

Accordingly, we present SIFT4G with Haplotype E as our primary predictor and reference, and Grantham scores together with Haplotype W as orthogonal checks for cross-species comparisons, with revisions added in the main text (Lines 252–274) and Supplementary Information (Figs. S29–S30). For within-species analyses (haplotype-level contrasts in *C. hunanensis* and embryo data from the three apomictic species), lineage divergence is not a confounding factor, so SIFT4G results based on Haplotype E were retained.

3. Inconsistent Interpretation of Results.

(a) It claims apomicts have "lower realized mutation loads" but acknowledges they harbor "more heterozygous deleterious variants." This tension is attributed to "hybrid buffering" in *C. hunanensis*, but no mechanistic evidence (e.g., gene expression data showing masked recessive effects) supports this assertion.

Answer: We agree that our study does not provide direct mechanistic evidence (e.g., gene expression data) for the masking of recessive deleterious alleles in the hybrid apomictic species *C. hunanensis*. We have therefore revised the Discussion (Lines 381–

387) to clarify this, adding the following text:

*“Among them, the hybrid apomict *C. hunanensis* is particularly notable, showing genomic patterns consistent with expectations under the hybrid buffering effect⁶⁴, whereby admixture between highly divergent parental genomes promotes fixed heterozygosity⁶⁵. While our observations align with this theoretical framework, we note that direct functional validation (e.g., gene expression assays) will be required to confirm the underlying mechanism.”*

(b) LOH is proposed to "expose deleterious mutations to selection," yet sexual embryos with LOH show higher homozygous deleterious loads. The suggestion that "selection eliminates harmful alleles early" is speculative and unsupported by data on embryo viability or fitness.

Answer: We thank the reviewer for this comment and acknowledge that our original wording was unclear. We have revised the sentence (Lines 410–417) to read:

*“While some sexually derived mature embryos showed elevated mutation loads, newly formed LOH regions in two such embryos—one from *C. cathayensis* and one from *C. dabieshanensis*—contained no homozygous LOF mutations. This pattern is suggestive of selection acting at multiple stages, with some genotypes carrying highly deleterious alleles possibly purged early, whereas other sexual embryos still retained homozygous LOF mutations, indicating that selection may also act later in development or during seedling establishment.”*

We agree that this remains indirect evidence. Direct confirmation would require analyses of fruits and seedlings at different developmental stages, which would provide a clearer picture of how LOH and deleterious alleles interact during embryogenesis.

Reviewer #2 (Remarks to the Author):

The manuscript uses advanced kinship analysis of *Carya* species with dense SNP data and the KING algorithm to study clonality and possible apomixis. The authors apply the KING algorithm, which allows them to analyze unfiltered SNP datasets and look for duplicated (clonal), first, second, and third-degree kin relationships. The study claims that there is a lot of clonality, sometimes almost complete, within these species.

By adding resequencing data from adult tissues and sequencing 180 embryos, the authors try to answer old questions about hybridization, persistent lineages, and the effect of apomictic versus sexual reproduction in this complicated hickory group. The study is relevant, the methodology is solid, and the evolutionary and ecological questions are important. Still, some points in the manuscript need to be explained better to make the conclusions clearer and reliable. Studying clonality on a genome-wide level in woody apomictic plants is new and important, including for the ecology and agriculture of hickory. The use of site frequency spectra (SFS) and loss-of-heterozygosity (LOH) analysis makes the argument for clonality even stronger and gives different views on the genetic structure. The analysis shows clonality at the population level very well. Including 180 mature embryos from three apomictic species is a very valuable part of the study.

Answer: We sincerely thank the reviewer for the positive evaluation of our study, particularly for recognizing the novelty and importance of applying genome-wide kinship, SFS, and LOH analyses in a clonal or asexual system, and for highlighting our efforts to address longstanding questions about the genomic consequences of rare sexual reproduction for lineage persistence in apomicts.

The authors say that rare loss-of-heterozygosity (LOH) events in embryos are probably caused by leftover recombination from occasional sexual reproduction, not from mitotic mistakes. This explanation is reasonable and interesting, and it fits with the finding that there is no genome-wide LOH. However, it would be helpful to better explain the difference between LOH caused by mitosis and LOH caused by automixis. For example, this could be done by showing allele imbalance of LOH patterns on chromosomes.

Answer: We thank the reviewer for this constructive suggestion. We agree that clarifying the mechanistic differences between LOH generated by mitotic repair, automixis, and residual sexual recombination improves interpretation.

Mitotic repair of DNA double-strand breaks typically generates very short and highly localized LOH tracts through gene conversion or break-induced replication—usually hundreds of bp to a few kb (Symington et al., 2014, *Genetics*). Such events are sporadic and locus-restricted, and LOH tracts are unlikely to be generated across multiple chromosomes.

Automixis, by contrast, produces genome-wide LOH because nearly all

chromosomes are affected. Crucially, these tracts follow structured and predictable patterns: terminal fusion yields LOH around centromeres, while central fusion generates LOH at chromosome ends while retaining heterozygosity near centromeres (Lenormand et al., 2016, *Genetics*). If automixis were the prevailing mechanism, such centromere- or telomere-anchored biases would recur consistently across embryos. However, our observed LOH tracts occur at variable chromosomal positions and do not match the expected automixis signatures.

Residual sexual recombination offers the best explanation: meiotic crossing-over during rare sexual events produces stochastic, embryo-specific LOH tracts that vary in location among individuals. This scenario fits our data from three apomictic *Carya* species, where 8/60, 11/60, and 3/60 embryos carried large LOH tracts.

As you suggested, we have added a schematic illustrating the characteristic LOH signatures expected under mitotic repair, automixis, and residual sexual recombination (Fig. S14), and we have revised the corresponding Results and Methods sections in the main text accordingly (Lines 200–232, 655–673).

Figure S14. Schematic comparison of loss-of-heterozygosity (LOH) signatures expected under different mechanisms.

A Mitotic repair typically produces very short, localized LOH tracts (blue) arising from gene conversion or break-induced replication, usually restricted to a few hundred bp to kb. **B** Automixis generates genome-wide LOH with structured patterns: terminal fusion yields LOH around centromeres, whereas central fusion produces LOH at chromosome ends while retaining heterozygosity near centromeres. **C** Residual sexual recombination results in stochastic LOH tracts of variable chromosomal positions across embryos, reflecting random meiotic crossing-over. Blue regions indicate LOH, grey regions indicate heterozygous segments.

My main doubts come from the fact that *Carya* is a sexually outcrossing (allogamous) species. Generally, in diploid species, the level of heterozygosity is roughly halved every generation when selfing occurs. In contrast, cross-pollination keeps or even

increases heterozygosity in the population, preserving genetic variation. In outcrossers, selfing causes higher LOH with a fast increase of homozygosity and a decrease of heterozygosity. Cross-pollination results in less LOH and keeps heterozygosity and genetic diversity high. This is why allogamous species that do repeated selfing experience a quick loss of heterozygosity and often suffer inbreeding depression. My personal idea to explain the results (embryos from sexual reproduction showing LOH), together with the authors' conclusion, is that in the crosses, both pollen and egg come from two plants of the same clone. But this should be studied more carefully. Still, seeing such LOH in only one generation (embryos from a mother plant) with crossing is unexpected.

Answer: We fully agree with the reviewer's point. In our *Carya* system, the sexual embryos showing LOH are most plausibly explained by selfing or by intra-clone/close-kin inbreeding during rare sexual events. This interpretation is supported by (i) an ~50% reduction in heterozygosity relative to apomictic mothers (Fig. S31), and (ii) the presence of first–third-degree relatedness among adults, which is compatible with mating within clonal lineages as well as inbreeding among closely related genets under the wind-pollinated condition. We have clarified this interpretation in the manuscript and revised the Results accordingly (Lines 208–213).

The authors should maybe sequence three biological replicates only for the samples that they think come from sexual reproduction, just to confirm that the LOH is consistent.

Answer: We agree that biological replication would be the most direct way to confirm that LOH patterns in sexual embryos are consistent and not sequencing artifacts. However, as the embryo sequencing was performed in 2023 and DNA extracts were not preserved at sufficient quality, additional replicates cannot be generated for this dataset.

Importantly, several independent lines of evidence argue against artifacts. All apomictic embryos showed fully consistent, heterozygous genomes, whereas the putative sexual embryos displayed large-scale LOH across nearly every chromosome. Given our stringent quality controls, these patterns are unlikely to result from sequencing errors or misidentification of LOH.

I also suggest explaining this part more clearly. This is important because the embryo genotypes, especially related to LOH and recombination, are a large part of the study's analysis and interpretation. So, this section should be better described better in the main methods or detailed more clearly in the Supplementary Information. Also, I suggest adding more discussion about other possible reasons, like mitotic recombination or technical problems. In conclusion, this paper gives valuable information about apomixis, clonality, and how hybrid species in long-living trees stay stable through time. It just needs some data/revisions, especially to explain the embryo part better.

Answer: We are grateful for the reviewer's concluding remarks highlighting the value of our study. We also thank the reviewer for the helpful suggestion regarding the LOH

analyses in sexual embryos. As noted above, we have clarified and elaborated these analyses, providing more detailed descriptions of the procedures used to minimize potential technical artifacts together with a clearer comparison of possible mechanisms (including mitotic and automixis recombination). These revisions have been incorporated into the main text and the Supplementary Information.